# Endothelial progeria induces adipose tissue senescence and impairs insulin sensitivity through senescence associated secretory phenotype

Agian Jeffilano Barinda [1,2], Koji Ikeda[1]*, Dhite Bayu Nugroho [1], Donytra Arby Wardhana[1], Naoto Sasaki[3], Sakiko Honda[4], Ryota Urata[4], Satoaki Matoba[4], Ken-ichi Hirata[5] & Noriaki Emoto[1,5]

Vascular senescence is thought to play a crucial role in an ageing-associated decline of organ functions; however, whether vascular senescence is causally implicated in age-related disease remains unclear. Here we show that endothelial cell (EC) senescence induces metabolic disorders through the senescence-associated secretory phenotype. Senescence-messaging secretomes from senescent ECs induced a senescence-like state and reduced insulin receptor substrate-1 in adipocytes, which thereby impaired insulin signaling. We generated EC-specific progeroid mice that overexpressed the dominant negative form of telomeric repeat-binding factor 2 under the control of the Tie2 promoter. EC-specific progeria impaired systemic metabolic health in mice in association with adipose tissue dysfunction even while consuming normal chow. Notably, shared circulation with EC-specific progeroid mice by parabiosis sufficiently transmitted the metabolic disorders into wild-type recipient mice. Our data provides direct evidence that EC senescence impairs systemic metabolic health, and thus establishes EC senescence as a bona fide risk for age-related metabolic disease.

[1] Laboratory of Clinical Pharmaceutical Science, Kobe Pharmaceutical University, 4-19-1 Motoyamakitamachi, Higashinada, Kobe 658-8558, Japan. [2] Department of Pharmacology and Therapeutic, Faculty of Medicine, Universitas Indonesia, Salemba Raya 6, Jakarta 10430, Indonesia. [3] Laboratory of Medical Pharmaceutics, Kobe Pharmaceutical University, 4-19-1 Motoyamakitamachi, Higashinada, Kobe 658-8558, Japan. [4] Department of Cardiology, Kyoto Prefectural University Graduate School of Medical Science, 465 Kajii, Kawaramachi-Hirokoji, Kyoto 602-8566, Japan. [5] Division of Cardiovascular Medicine, Department of Internal Medicine, Kobe University Graduate School of Medicine, 7-5-1 Kusunoki, Chuo, Kobe 6500017, Japan. *email: ikedak-circ@umin.ac.jp

Advanced age is a significant risk factor for prevalent diseases such as diabetes and cardiovascular disease[1–4]. The vascular system consists of blood vessels that provide routes for circulation and transportation of nutrients, oxygen, and hormones throughout the body. The endothelial cell (EC) lines the inner surface of the blood vessels, and plays an essential role in vascular biology, such as vasodilation, hormone trafficking, and neovessel formation[5]. Moreover, EC produces many secreted angiocrine factors that are crucially involved in maintaining tissue homeostasis[6–9]. Aging causes cellular senescence in various types of cells including EC, and cellular senescence plays an important role in age-related organ dysfunction[10–13]. Senescent cells produce senescence-messaging secretomes that have deleterious effects on the tissue microenvironment, referred as the senescence-associated secretory phenotype (SASP)[14–16]; therefore, cellular senescence is considered to be a primary cause for age-related diseases, such as diabetes, stroke, and heart attack[2,17]. Because of the crucial roles of EC in tissue homeostasis[6–9], EC senescence is presumed to play significant roles in age-related organ dysfunction; however, whether and the mechanism by which EC senescence causes age-related diseases remained unknown.

We have previously reported that adipose tissue (AT) vasculatures are critically involved in fat functions, and consequently play crucial roles in the maintenance of systemic metabolic health[18–20]. Adipocytes actively regulate AT angiogenesis by secreting angiogenic factors to maintain AT vasculatures[19–23]. Impaired AT angiogenesis regulation and/or disproportional adipocyte hypertrophy leads to reduction in AT vasculature, resulting in fat dysfunction. Since blood vessels are the route for oxygen and nutrition transport, loss of AT vasculatures causes hypoxia and nutritional disorders that might impair fat functions. We also presume that ECs play some roles in the maintenance of fat functions through angiocrine factors, and thus not only quantity but also the quality of EC might be important for AT health. We, therefore, hypothesized that degraded quality of EC during aging may actively impair fat functions, leading to the age-related metabolic disorders. Here, we explore a potential role of EC senescence in age-related metabolic disorders by utilizing newly generated EC-specific progeroid mice, and identify deleterious effects of endothelial senescence-messaging secretomes on the maintenance of adipocyte functions and systemic metabolic health.

## Results

### Senescent EC impairs adipocyte functions through SASP.
Aging causes impaired metabolic homeostasis[1,2,24], and the AT contains highly developed vascular networks that play an important role in maintaining adipocyte functions[18,21,22]. We, therefore, hypothesized that EC senescence might directly cause the adipocyte dysfunction and lead to age-related metabolic disorders. To analyze the role of EC senescence, we prepared two types of senescent EC: replicative senescent and premature senescent ECs by using human umbilical vein ECs. Premature senescence was induced by overexpressing the dominant negative (DN) form of the telomeric repeat-binding factor 2 (TERF2)[25,26] in ECs. Their senescent phenotypes were validated through reduced proliferation, DNA damage, senescence-associated (SA) heterochromatin foci, reduced histone H3 lysine 9 dimethylation, SA β-galactosidase (Gal) activity, and increased CDK inhibitor and SASP factor expression (Supplementary Fig. 1a–g). We also confirmed that overexpression of TERF2DN did not induce EC apoptosis in the absence of a cytotoxic stress (Supplementary Fig. 1h). Replicative senescence of ECs was also validated through DNA damage, SA heterochromatin foci, and increased SASP

factor expression (Supplementary Fig. 2) in addition to the reduced proliferation and increased CDK inhibitor expression, which we reported previously[26]. When treated with replicative senescent EC-conditioned medium (CM) that is enriched with endothelial senescence-messaging secretomes, 3T3-L1 adipocytes exhibited multiple senescence-like features such as SA-β-Gal activity, and increased CDK inhibitor and SASP factor expression compared with cells treated with control medium or CM derived from proliferating young EC (Fig. 1a–c). Although adipocytes are terminally differentiated nondividing cells, some of post-mitotic cells, including neurons and adipocytes, have been reported to exhibit several SA properties, and thus post-mitotic senescence-like state in terminally differentiated cells is an emerging concept[27,28]. Consistently, white adipose tissue (WAT) isolated from aged mice showed increased CDK inhibitor expression compared with that in WAT of young mice (Supplementary Fig. 3a). Of note, these 3T3-L1 adipocytes in senescence-like state showed impaired insulin signaling in association with reduced insulin receptor substrate (IRS)-1 expression (Fig. 1d–f). Similar results were obtained when 3T3-L1 adipocytes were treated with CM derived from premature senescent EC (Fig. 2). In contrast, treatment with senescent EC-CM did not induce such senescence features in 3T3-L1 preadipocytes or C2C12 myotubes (Supplementary Fig. 3b–d). In addition, we explored the paracrine effects of endothelial SASP on EC. Treatment with senescent EC-CM did not induce senescence features in young proliferating EC (Supplementary Fig. 4). These data suggest a higher susceptibility of mature adipocytes toward the endothelial SASP.

### Senescent EC induces oxidative stress in adipocytes.
Oxidative stress is closely associated with cellular senescence. We found that treatment with senescent EC-CM caused excessive superoxide production in 3T3-L1 adipocytes, suggesting that endothelial senescence-messaging secretomes induced oxidative stress (Fig. 3a, b). Of note, treatment with the antioxidant N-acetylcystein (NAC) prevented endothelial SASP-mediated senescence features and impaired insulin signaling in adipocytes as well as β-nicotinamide mononucleotide, an intermediate of $NAD^+$ biosynthesis that activates sirtuin[29] (Fig. 3c–e). Pharmacological inhibition of ROS production pathways, such as xanthine and NADPH oxidase, failed to reverse senescence features in adipocytes treated with senescent EC-CM (Supplementary Fig. 5a), whereas expression of superoxide dismutase was reduced in these cells (Supplementary Fig. 5b). These data collectively suggest that senescent EC induces senescence-like state and impairs insulin signaling in adipocytes through the SASP by enhancing oxidative stress due to a defective redox state.

### Generation of EC-specific progeroid mice.
To analyze a role of senescent EC in age-related fat dysfunction in vivo, separately from effects of cellular senescence of other types of cells, we generated EC-specific progeroid mice that overexpress the DN form of TERF2 under the control of the Tie2 promoter (Tie2-TERF2DN-Tg). We obtained two lines of transgenic mice, both of which were viable and fertile. Senescence features, such as increased CDK inhibitor and SASP factor expressions, were detected in ECs isolated from Tie2-TERF2DN-Tg mice, while non-ECs did not show such features (Figs. 4a, b, Supplementary Fig. 6a, b). SA-β-Gal activity was also detected in ECs but not in non-ECs isolated from Tie2-TERF2DN-Tg mice (Fig. 4c). The percentage of SA-β-Gal-positive ECs in Tie2-TERF2DN-Tg mice was comparable with that in ECs isolated from naturally aged mice (Fig. 4d). We then explored whether these premature senescent ECs in Tie2-TERF2DN-Tg mice resemble senescent ECs in naturally aged mice. Principal component analysis for

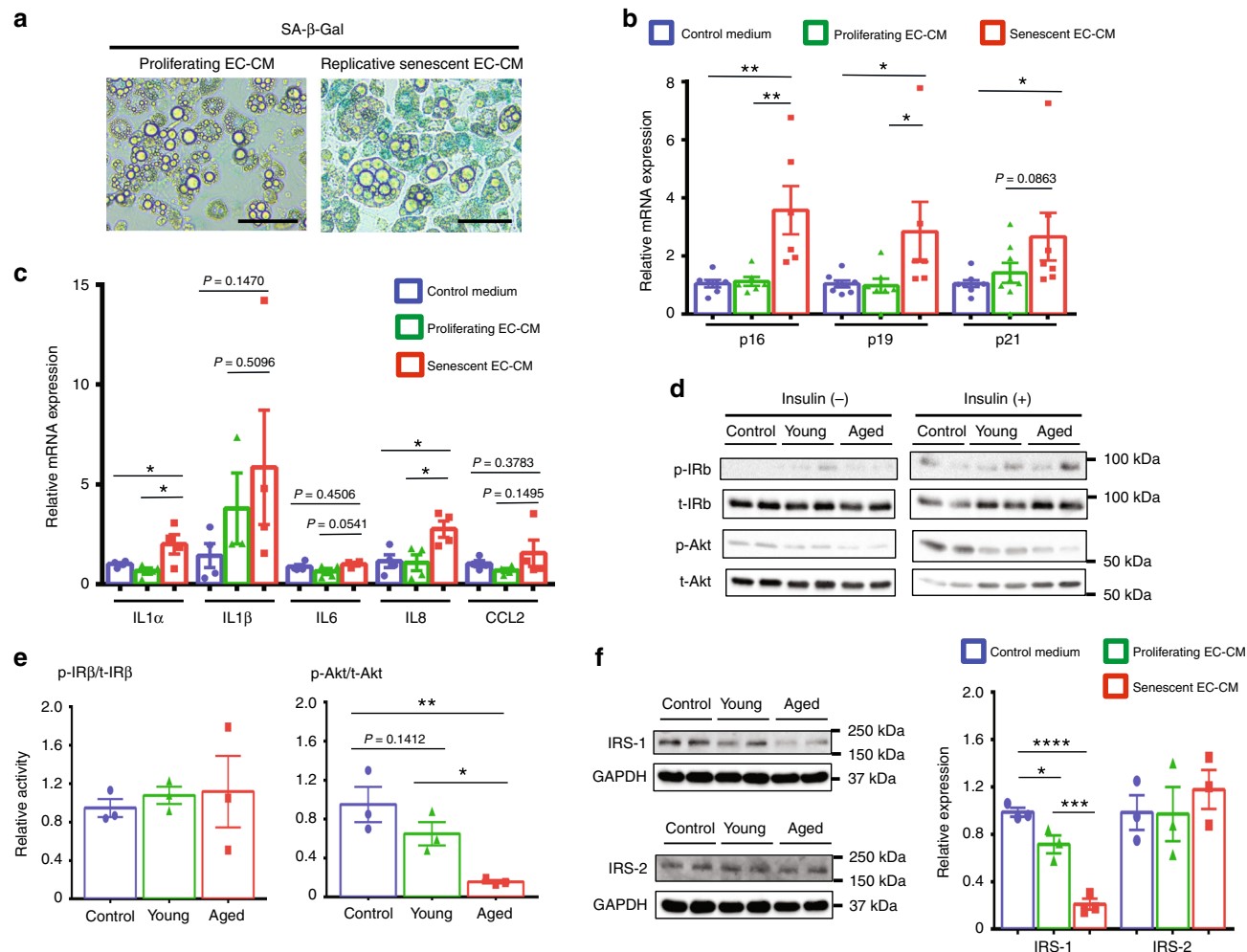

**Fig. 1 Replicative senescent EC impairs adipocyte function through the SASP. a** SA-β-Gal staining of 3T3-L1 adipocytes treated with the conditioned medium (CM) derived from proliferating young or replicative senescent EC. **b** CDK inhibitor expression in 3T3-L1 adipocytes treated with the control medium, or CM derived from proliferating young or replicative senescent EC ($n = 8$ biologically independent samples for control medium group; $n = 6$ biologically independent samples for EC-CM groups). **c** SASP factor expression in 3T3-L1 adipocytes treated with the control medium, or CM derived from proliferating young or replicative senescent EC ($n = 4$ biologically independent samples each). **d** Immunoblotting for the insulin signal pathway in 3T3-L1 adipocytes treated with the control medium (control), CM derived from proliferating young EC (young), or CM derived from replicative senescent EC (aged) in the presence or absence of insulin treatment. **e** Quantitative analysis for IRβ and Akt activation in response to insulin ($n = 3$ biologically independent samples each). **f** Immunoblotting for IRS-1 and IRS-2 in 3T3-L1 adipocytes treated with the control medium, or indicated EC-CM. Quantitative analysis was also shown ($n = 3$ biologically independent samples each). Non-repeated ANOVA with post hoc analysis of Fisher's PLSD was used for difference evaluation between the groups (**b**, **c**, **e**, **f**). Data are presented as mean ± s.e. *$P < 0.05$, **$P < 0.01$, ***$P < 0.001$, and ****$P < 0.0001$. Bars: 100 μm. Source data are provided as a Source Data file.

gene expression profiles assessed by DNA microarray revealed a significant similarity between ECs isolated from Tie2-TERF2DN-Tg and naturally aged mice, which significantly differed from ECs isolated from young mice (Fig. 4e). These data collectively indicate the EC-specific progeria in Tie2-TERF2DN-Tg mice, and provide a rationale for using this mouse model to analyze a role of EC senescence in aging.

**EC senescence impairs systemic insulin sensitivity in vivo.** Body weight was similar between the Tie2-TERF2DN-Tg and WT mice, although there was a trend toward an increase in the Tie2-TERF2DN-Tg mice (Fig. 5a). Body weight and body fat ratio at the age of 20-week old also showed a tendency toward an increase in the Tie2-TERF2DN-Tg mice, while the differences did not reach statistical significance (Fig. 5b, c). Of note, Tie2-TERF2DN-Tg mice exhibited impaired metabolic health at, as early as, 20-week old even while consuming normal chow (Fig. 5d, Supplementary

Fig. 8a). Consistent with the reduced insulin sensitivity, serum insulin levels were higher in the Tie2-TERF2DN-Tg mice than those in the WT mice, whereas serum lipid profiles were similar across groups (Supplementary Figs. 7a and 8b). Because the insulin tolerance curve appeared to diverge especially at a later time point, we examined a possibly enhanced counter-regulatory response in the Tie2-TERF2DN-Tg mice through glucocorticoids. Serum corticosterone levels were similar between the WT and Tie2-TERF2DN-Tg mice (Supplementary Fig. 7b). On the other hand, CDK inhibitor expression increased not only in the EC-rich stromal vascular fraction (Supplementary Fig. 7c), but also in the mature adipocytes isolated from the WAT of Tie2-TERF2DN-Tg mice (Fig. 5e). Consistently, significant SA-β-Gal activity was detected in the WAT of Tie2-TERF2DN-Tg mice (Fig. 5f). In addition, some of inflammatory gene expression was increased in the WAT of Tie2-TERF2DN-Tg mice (Supplementary Fig. 8c). Furthermore, insulin signaling was impaired in the WAT of

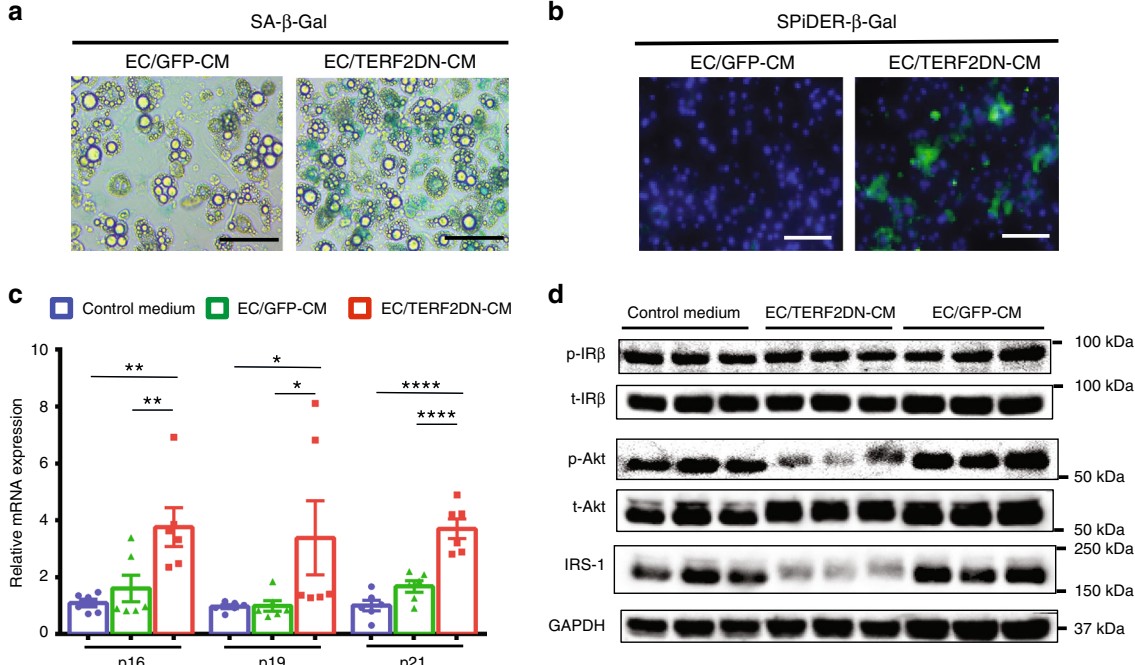

**Fig. 2 Premature senescent EC impairs adipocyte function through the SASP. a** SA-β-Gal staining in 3T3-L1 adipocytes treated with CM derived from control ECs (EC/GFP-CM) or premature senescent ECs (EC/TERF2DN-CM). **b** SPiDER-β-Gal staining in 3T3-L1 adipocytes treated with CM derived from control ECs (EC/GFP-CM) or premature senescent ECs (EC/TERF2DN-CM). **c** CDK inhibitor expression in 3T3-L1 adipocytes treated with the control medium, EC/GFP-CM or EC/TERF2DN-CM ($n = 6$ biologically independent samples each). **d** Immunoblotting for the insulin signal pathway and IRS-1 in insulin-stimulated 3T3-L1 adipocytes treated with the control medium, EC/GFP-CM or EC/TERF2DN-CM. Non-repeated ANOVA with post hoc analysis of Fisher's PLSD was used for statistical analysis. Data are presented as mean ± s.e. *$P < 0.05$, **$P < 0.01$, and ****$P < 0.0001$. Bars: 100 μm. Source data are provided as a Source Data file.

Tie2-TERF2DN-Tg mice in association with reduced IRS-1 expression as compared with that in WT mice (Fig. 5g, h). These results indicate that EC senescence is sufficient to induce senescence-like state and dysfunction in adipocytes in vivo, which probably have enough deleterious impact on systemic metabolic health. Notably, neither adipocyte senescence-like state nor metabolic disorders were observed in 10-week-old Tie2-TERF2DN-Tg mice (Fig. 6a, b), though ECs already showed senescence features at this age (Fig. 6c, d). These results indicate that WAT dysfunction and metabolic disorders in Tie2-TERF2DN-Tg mice occur postnatally after EC senescence, and are not due to developmental defects of the WAT. Blood vessel density in white and brown AT did not differ between the Tie2-TERF2DN-Tg and WT mice (Supplementary Fig. 8d, e). Similar phenotypes, including the EC senescence, impaired insulin sensitivity, and senescence-like state in WAT, were detected in an independent line of Tie2-TERF2DN-Tg mice at the age of 20 weeks old (Supplementary Fig. 9).

**WAT is highly susceptible to EC-mediated SASP.** In contrast to WAT, there were no significant changes in CDK inhibitor expression in other tissues, including the liver, skeletal muscle, and BAT (Supplementary Fig. 10a–c). Gluconeogenic gene expression in the liver, PGC-1α expression in the skeletal muscle, thermogenic gene expression in the brown AT, and core temperature were not altered in Tie2-TERF2DN-Tg mice (Supplementary Fig. 10d–g). Moreover, insulin signaling was not impaired in the liver or skeletal muscle of Tie2-TERF2DN-Tg mice, in contrast to WAT (Fig. 5h). These data strongly suggest that WAT is more susceptible to the endothelial SASP than other organs as suggested by the in vitro studies.

**Oxidative stress in WAT of EC-specific progeroid mice.** WAT isolated from Tie2-TERF2DN-Tg mice showed significant oxidative DNA damage assessed by immunostaining for 8-hydroxy-2′-deoxyguanosine (8-OHdG) (Fig. 7a). We, therefore, explored whether enhanced oxidative stress plays a causative role in the impaired metabolic health in EC-specific progeroid mice. Administration of antioxidants for 10 weeks beginning at the age of 10 weeks old prevented the development of insulin resistance and WAT dysfunction in Tie2-TERF2DN-Tg mice (Fig. 7b–e). These data strongly suggest that oxidative stress induced by EC senescence plays a causative role in the WAT dysfunction and impaired systemic metabolic health in EC-specific progeroid mice.

**EC senescence is sufficient to impair metabolic health.** To exclude the possible role of bone marrow (BM) cells that potentially express TERF2DN under the control of the Tie2 promoter, we generated BM chimeric mice (Supplementary Fig. 11). Tie2-TERF2DN-Tg mice transplanted with WT-BM cells remained to be insulin resistant, accompanied by the WAT senescence features, whereas WT mice transplanted with Tie2-TERF2DN-Tg-BM showed normal insulin sensitivity (Fig. 8a, b). These data indicate the minimal contribution of BM cells for the impaired metabolic health in Tie2-TERF2DN-Tg mice, and further validate a causative and crucial role of EC senescence in age-related metabolic disorders.

**EC progeria impairs metabolic health through the SASP.** To investigate whether systemic metabolic disorders are induced by soluble factors in the blood circulation in EC-specific progeroid mice, we generated WT mice in which blood circulation was

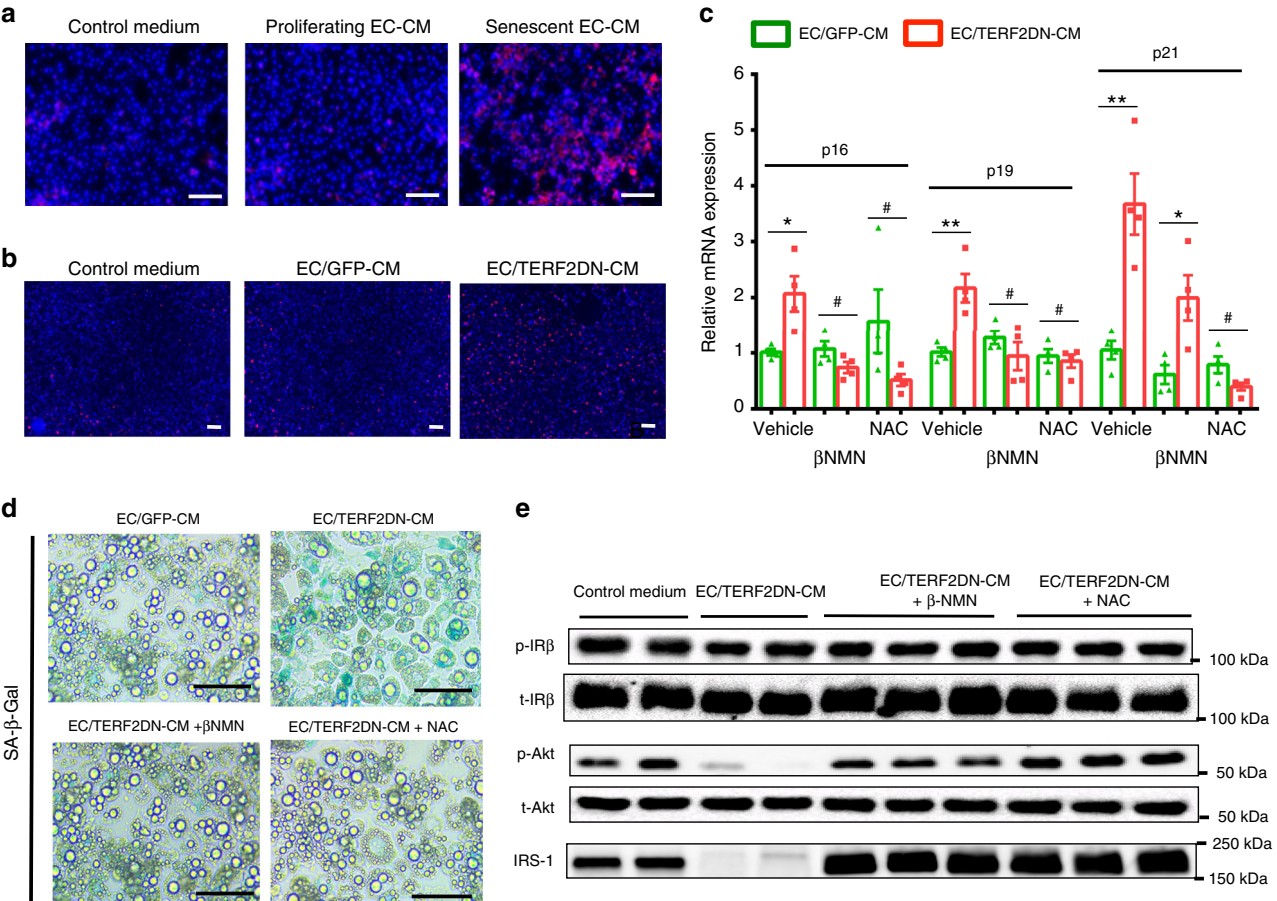

**Fig. 3 Senescent EC impairs adipocyte functions by inducing oxidative stress. a** Superoxide was detected (as red fluorescence) in 3T3-L1 adipocytes treated with the control medium, or CM derived from proliferating young or replicative senescent EC. **b** Superoxide was detected in 3T3-L1 adipocytes treated with the control medium, or CM derived from control ECs (EC/GFP-CM) or premature senescent ECs (EC/TERF2DN-CM). **c** CDK inhibitor expression in 3T3-L1 adipocytes treated with the indicated EC-CM in the presence or absence of βNMN or NAC ($n = 4$ biologically independent samples each). A two-tailed Student's $t$ test was used for difference evaluation between the two groups. Data are presented as mean ± s.e. *$P < 0.05$, **$P < 0.01$, and #not significant. **d** SA-β-Gal staining in 3T3-L1 adipocytes treated with the indicated EC-CM in the presence or absence of βNMN or NAC. **e** Immunoblotting for the insulin signal pathway and IRS-1 in insulin-stimulated 3T3-L1 adipocytes treated with the indicated EC-CM in the presence or absence of βNMN or NAC. Bars: 100 μm. Source data are provided as a Source Data file.

shared with Tie2-TERF2DN-Tg or littermate WT mice by using parabiosis procedure[30]. The surgical procedures for parabiosis were performed by using 10-week-old recipient WT and donor Tie2-TERF2DN-Tg or WT mice, and phenotypic analysis was performed after a 10-week period of shared circulation. Recipient WT mice whose circulation was shared with Tie2-TERF2DN-Tg mice (WT-Tg) showed significantly impaired insulin sensitivity compared with that in control recipient mice (WT–WT) to the extent similar to that in recipient Tg mice with shared circulation with Tg mice (Tg–Tg) (Fig. 8c). These data strongly suggest that senescence-messaging secretomes produced by senescent EC in blood circulation impair systemic metabolic health in an endocrine-dependent manner. We also explored whether blood from WT mice affects the metabolic disorders in Tie2-TERF2DN-Tg mice, and found that shared circulation with WT mice partly reversed the impaired insulin sensitivity in Tie2-TERF2DN-Tg mice (Fig. 8d).

**IL-1a orchestrates the cytokine networks in senescent EC.** Interleukin (IL)-1a has been reported as a marker of senescent but not quiescent ECs[31]. Furthermore, cell-surface-bound IL-1a has been reported to regulate senescence-associated cytokine

networks in fibroblasts[32]. We also found that IL-1a showed a remarkable increase in replicative and premature senescent ECs in vitro, as well as in ECs isolated from Tie2-TERF2DN-Tg mice in vivo. Therefore, we explored a role of IL-1a in the endothelial SASP. Despite the high mRNA expression, IL-1a was hardly detectable in the culture medium of replicative senescent EC (Fig. 9a). It has been reported that inflammasome activation is needed to secrete IL-1a from cells, while membrane-bound IL-1a is also biologically active[33,34]. We, therefore, treated senescent EC with inflammasome activators, and found that IL-1a was abundantly secreted into the culture medium of replicative senescent EC with these stimuli (Fig. 9a). Moreover, FACS analysis revealed that the membrane-bound IL-1a increased in ECs isolated from naturally aged mice compared with that in ECs isolated from young mice (Fig. 9b). We then investigated the role of membrane-bound IL-1a in the senescence-messaging secretomes in EC. Treatment with IL-1 receptor antagonist abolished the increase of multiple cytokines in senescent EC compared with those in proliferating young EC (Fig. 9c). Notably, conditioned medium derived from senescent EC that was treated with IL-1 receptor antagonist failed to induce senescence-like state in 3T3-L1 adipocytes (Fig. 9d). Because IL-1 receptor antagonist inhibits both IL-1a and IL-1b, we examined the effect of IL-1a gene silencing by

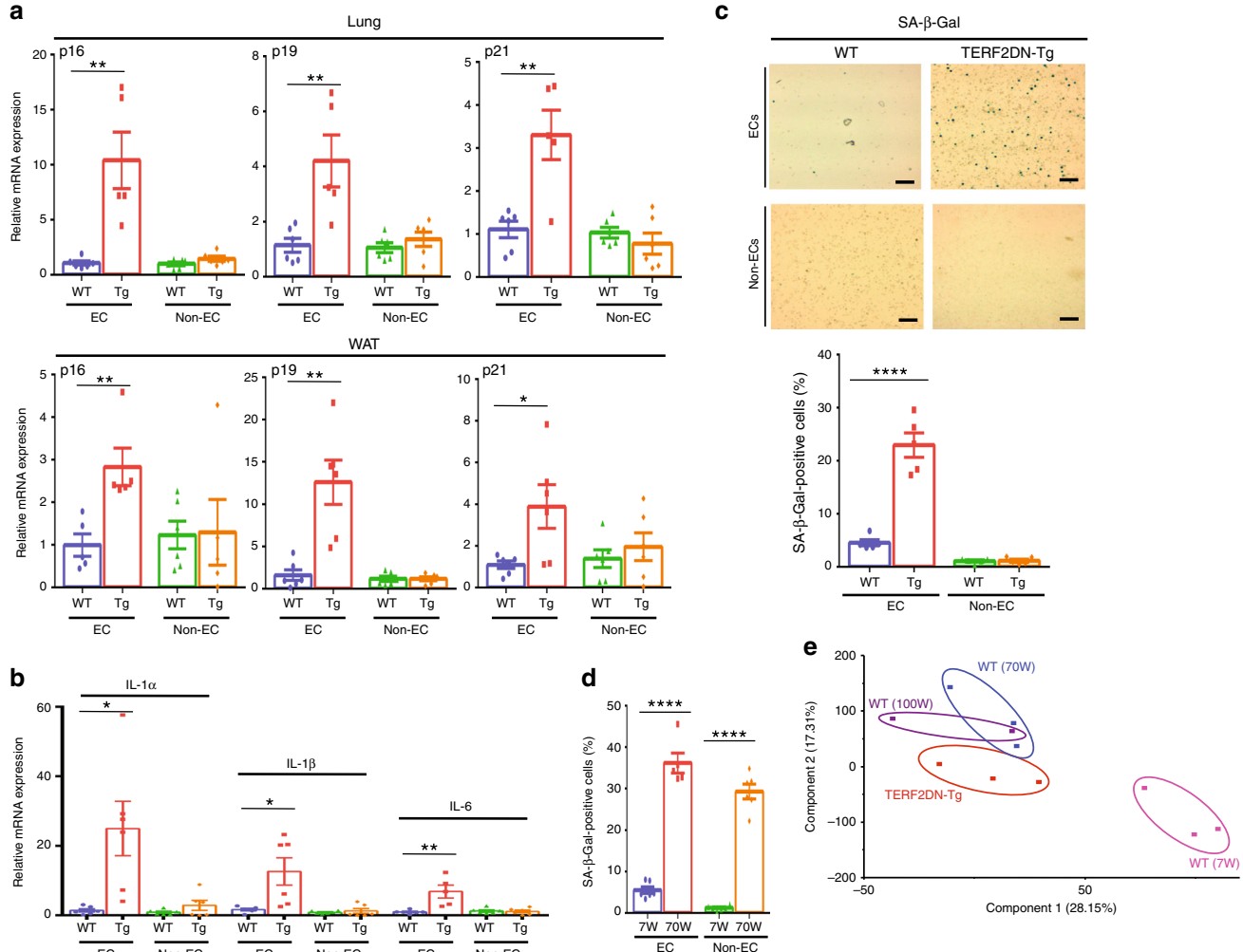

**Fig. 4 Generation of EC-specific progeroid mice. a** CDK inhibitor expression in EC and non-EC isolated from the lung and WAT of WT or Tie2-TERF2DN-Tg mice (line #16) ($n = 6$ biologically independent samples for WT; $n = 5$ biologically independent samples for Tg). **b** SASP factor expression in EC and non-EC isolated from the WAT of WT or Tie2-TERF2DN-Tg mice (line #16) ($n = 6$ biologically independent samples each). **c** SA-β-Gal staining in EC and non-EC isolated from the lung of WT or Tie2-TERF2DN-Tg mice (line #16). SA-β-Gal-positive cells were counted ($n = 5$ biologically independent samples each). **d** EC and non-EC were isolated from the lung of young (7W) or aged (70W) mice. Cells were stained with SA-β-Gal, and staining-positive cells were counted ($n = 6$ biologically independent samples each). **e** Principalcomponent analysis for gene expression profiles assessed by DNA microarray in ECs isolated from the lung of young WT (7W), aged WT (70W), or Tie2-TERF2DN-Tg mice (20-week old). A two-tailed Student's $t$ test was used for difference evaluation between the two groups (**a–d**). Data are presented as mean ± s.e. *$P < 0.05$, **$P < 0.01$, and ****$P < 0.0001$. Bars: 100 μm. Source data are provided as a Source Data file.

using short-interfering RNA. Similar to IL-1 receptor antagonist, gene silencing of IL-1a abrogated the increase of multiple cytokines in senescent EC compared with those in control EC, without affecting the CDK inhibitor expression (Fig. 9e, f). These data strongly suggest a critical role of membrane-bound IL-1a in the endothelial SASP by orchestrating the senescence-associated cytokine networks in EC.

**Analysis for EC secretomes.** Characterization of changes in EC secretomes during aging provides invaluable information to identify SASP factors that mediate adipocyte dysfunction. We, therefore, performed the shotgun proteomics analysis using CM derived from young control and premature senescent EC. We detected 1307 proteins in the CM (Supplementary Excel sheet). Among these proteins, 380 proteins were differentially regulated ($P < 0.1$) in the senescent EC-CM compared with control EC-CM (Supplementary Fig. 12). Pathway analysis for the proteins that are increased in senescent EC-CM revealed several pathways that

are potentially affected by the endothelial SASP, including signaling pathway for insulin-like growth factor, basic fibroblast growth factor platelet-derived growth factor, and extracellular matrix organization (Supplementary Fig. 13).

## Discussion

In this study, we revealed a deleterious impact of EC senescence on adipocyte function and systemic metabolic health by utilizing newly generated EC-specific progeroid mice. Elimination of p16-positive senescent cells delayed SA disorders, such as atherosclerosis, and notably extended the median lifespan in mice[35–37]. Also, targeted apoptosis of senescent cells, by using FOXO4 peptide, restored fitness, fur density, and renal function in aged mice[38]. Moreover, senolytic treatments that selectively kill senescent cells improved vasomotor function and reduced aortic calcification in aged mice[39,40]. Therefore, cellular senescence could be a primary cause for age-related diseases. ECs line the inner surface of blood vessels that are extended to all the organs,

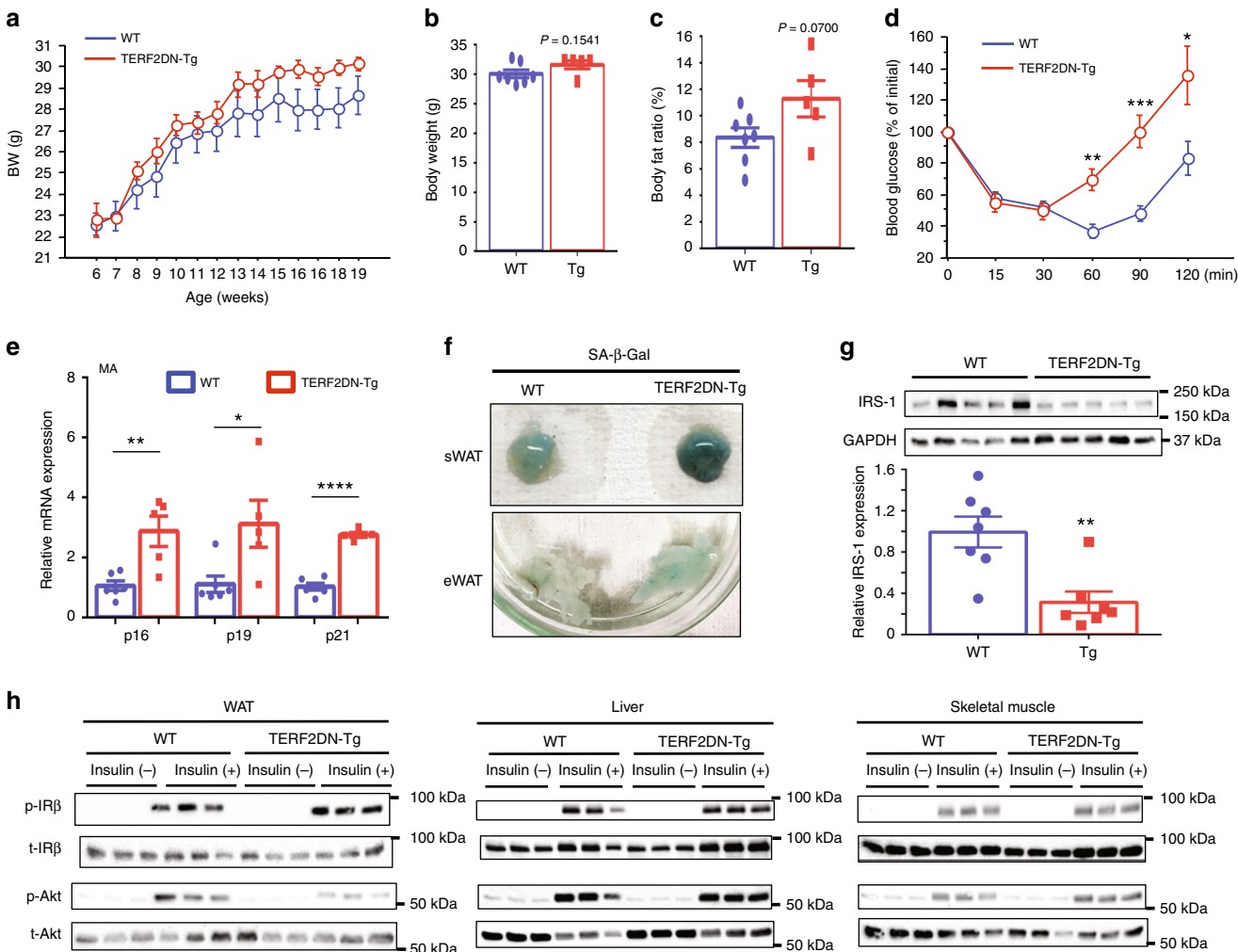

**Fig. 5 EC-specific progeria impairs systemic metabolic health. a** Body weight of WT or Tie2-TERF2DN-Tg (line #16) mice fed normal chow (NC) ($n = 7$ biologically independent animals for WT; $n = 6$ biologically independent animals for Tg). **b**, **c** Body weight (**b**) and body fat ratio (**c**) in WT or Tie2-TERF2DN-Tg mice fed NC at the age of 20 weeks old ($n = 7$ biologically independent animals for WT; $n = 5$ biologically independent animals for Tg). **d** Insulin tolerance test (ITT) in WT or Tie2-TERF2DN-Tg mice fed NC at the age of 20 weeks old ($n = 7$ biologically independent animals for WT; $n = 6$ biologically independent animals for Tg). **e** CDK inhibitor expression in mature adipocytes (MA) isolated from the WAT of WT or Tie2-TERF2DN-Tg mice ($n = 6$ biologically independent samples for WT; $n = 5$ biologically independent samples for Tg). **f** SA-β-Gal staining of subcutaneous (sWAT) or visceral epididymal WAT (eWAT) isolated from WT or Tie2-TERF2DN-Tg mice. **g** Immunoblotting for IRS-1 in the WAT isolated from WT or Tie2-TERF2DN-Tg mice ($n = 7$ biologically independent samples each). **h** Immunoblotting for the insulin signal pathway in the WAT, liver, and skeletal muscle of NC-fed WT or Tie2-TERF2DN-Tg mice with or without insulin treatment. A two-tailed Student's $t$ test was used for difference evaluation between the two groups (**b–e**, **g**). Data are presented as mean ± s.e. *$P < 0.05$, **$P < 0.01$, ***$P < 0.001$, and ****$P < 0.0001$. Source data are provided as a Source Data file.

and thus ECs are unique and the most abundant cellular compositions in the body[5,41]. Accordingly, EC senescence has been presumed to play particular roles in aging in addition to altering blood vessel functions; however, whether EC senescence is causally implicated in age-related disease had remained unclear. Our in vitro and in vivo studies presented here revealed that EC senescence directly caused adipocyte dysfunction and consequently impaired systemic metabolic health, indicating a causative role of EC senescence in age-related metabolic disorders.

Senescent cells develop a persistent pro-inflammatory phenotype, called SASP that deteriorates the microenvironment and impairs cellular functions in nearby cells[16]. Our in vitro experiments strongly suggest that senescent EC impairs adipocyte functions through the senescence-messaging secretomes, although there is still a possibility that enhanced consumption of trace elements in the culture medium by senescent EC may play a role in adipocyte dysfunction treated with senescent EC-CM.

Moreover, the parabiosis experiments clearly showed that EC senescence impaired systemic metabolic health largely through factors in blood circulation. Therefore, SASP plays a central role in metabolic disorders caused by EC senescence. As we demonstrated a critical role of IL-1a in endothelial SASP in vitro, experiments by using IL-1a-knockout background mice will provide invaluable information to elucidate a role of endothelial SASP in vivo. Senescent EC-CM and blood in EC-specific progeroid mice contain not only soluble factors secreted from EC, but also microparticles, resulting from endothelial plasma membrane blebbing[42]. Microparticles carry many various proteins and nuclear materials, including DNA, RNA, and microRNA, which affect cellular functions[42]. Therefore, the mechanisms underlying the EC-mediated SASP are quite a challenge to understand. To obtain some clues for identification of endothelial SASP factors responsible for inducing senescence of AT, we performed the shotgun proteomics analysis by using the CM derived from young control

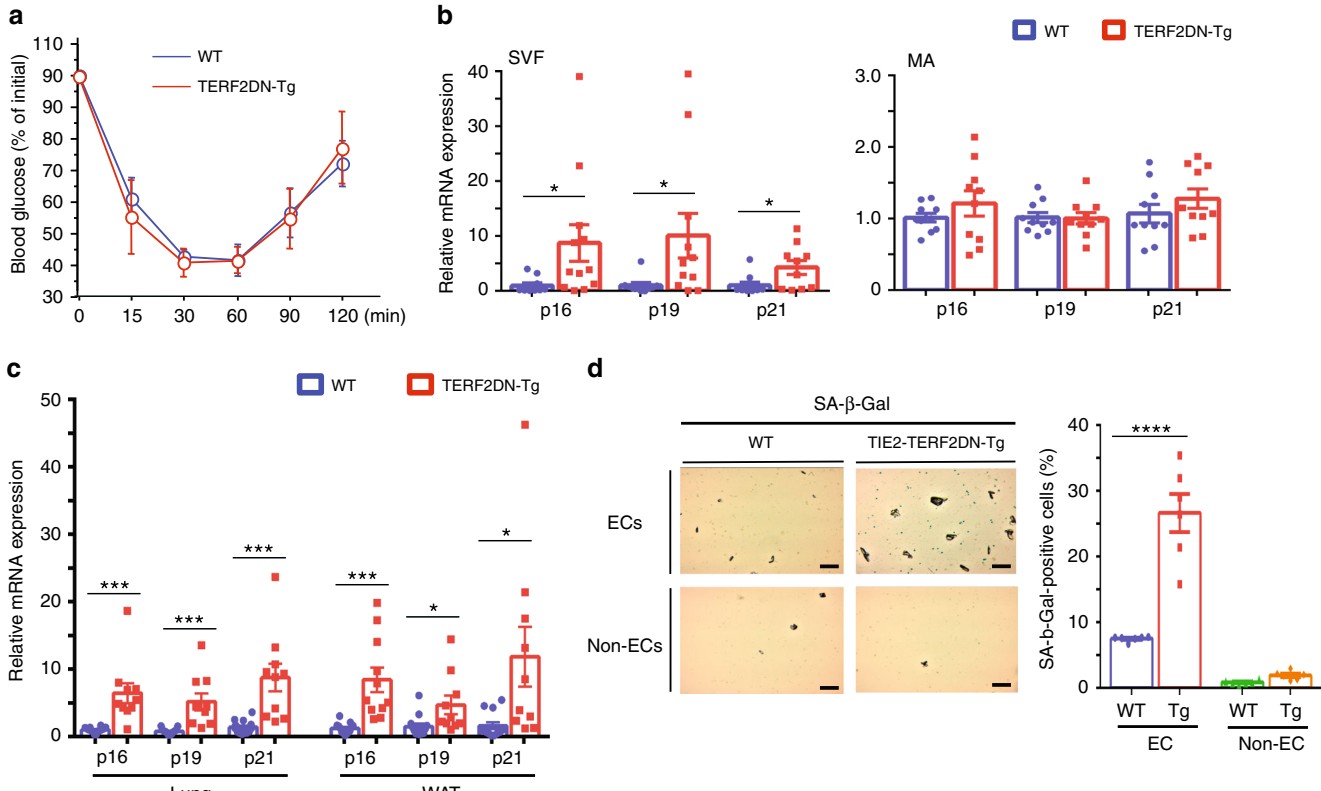

**Fig. 6 EC-specific progeroid mice are metabolically healthy at the age of 10-week old. a** ITT in WT or Tie2-TRF2DN-Tg mice fed normal chow at the age of 10-week old ($n = 7$ biologically independent animals for WT; $n = 5$ biologically independent animals for Tg). **b** CDK inhibitor expression in stromal vascular fraction (SVF) or mature adipocytes (MA) isolated from WAT of WT or Tie2-TRF2DN-Tg mice at the age of 10-week old ($n = 10$ biologically independent samples each). **c** CDK inhibitor expression in ECs isolated from lung ($n = 15$ biologically independent samples for WT; $n = 10$ biologically independent samples for Tg), and WAT ($n = 13$ biologically independent samples for WT; $n = 10$ biologically independent samples for Tg) of WT or Tie2-TRF2DN-Tg mice at the age of 10-week old. **d** SA-β-Gal staining in EC and non-EC isolated from the lung of WT or Tie2-TRF2DN-Tg mice at the age of 10-week old. Bars: 100 μm. SA-β-Gal-positive cells were counted ($n = 6$ independent fields each). A two-tailed Student's $t$ test was used for difference evaluation between the two groups (**b–d**). Data are presented as mean ± s.e. *$P < 0.05$, ***$P < 0.001$, and ****$P < 0.0001$. Source data are provided as a Source Data file.

and senescent ECs. We have identified several pathways that are potentially modified by the endothelial SASP. These pathways include the signaling pathway for insulin-like growth factor, basic fibroblast growth factor, and platelet-derived growth factor, which has been involved in cellular senescence and oxidative stress[43–47]. However, a significant number of proteins detected in the CM were not secreted but localized intracellularly. We presume that some of them might be actively secreted through vesicles, and some of them might be derived from dead cells. Moreover, many cytokines, including IL-1a, IL-1b, IL-6, IL-8, MCP-1, TNF-a, and TGF-b, were not detected, probably because of technical limitations. Because of these limitations, the whole picture of endothelial SASP has not yet been revealed, and thus it is difficult to determine SASP factor(s) responsible for inducing senescence of adipocytes at this point. Further analyses to obtain complementary data are certainly required to identify the causative factor(s) produced by senescent EC that impairs adipocyte function and metabolic health.

Regarding the molecular mechanism by which adipocyte senescence provokes systemic metabolic disorder, we presume that impaired insulin signaling in WAT plays a primary role in the impaired metabolic health in EC-specific progeroid mice. We have revealed that insulin signaling in WAT was impaired by senescent ECs in association with senescence-like features and IRS-1 reduction both in vitro and in vivo. Many previous papers

reported a critical role of WAT insulin signaling in the regulation of systemic metabolic homeostasis[48,49]. Furthermore, it has been reported that enhancing insulin signaling in adipocytes is sufficient to improve systemic metabolic homeostasis, while impaired insulin signaling in adipocytes causes systemic metabolic disorders[50,51]. Therefore, adipocyte senescence induces systemic metabolic disorders at least partially through impairing fat insulin signaling. We also found that WAT is more susceptible to endothelial SASP than other tissues, while the underlying mechanisms remain to be elucidated.

We found a substantial increase in IL-1a expression in senescent ECs both in vitro and in vivo, and our in vitro studies revealed a critical role of membrane-bound IL-1a in orchestrating the senescence-associated cytokine networks in EC, leading to the endothelial SASP for adipocytes. Therefore, inhibition of IL-1a has a therapeutic potential for the prevention and/or treatment of metabolic disorders in elderly population. Also, it has been reported that oxidative stress induced senescence-like features and impaired insulin signaling in adipocytes[52,53]. Consistently, we demonstrated that oxidative stress is primarily involved in the metabolic disorders in EC-specific progeroid mice. These data may suggest a theoretical rationale for antioxidant therapy to prevent age-related metabolic disease, though the benefits of antioxidant supplementation remain controversial[54]. Taken together, our study revealed that EC senescence is a bona fide risk

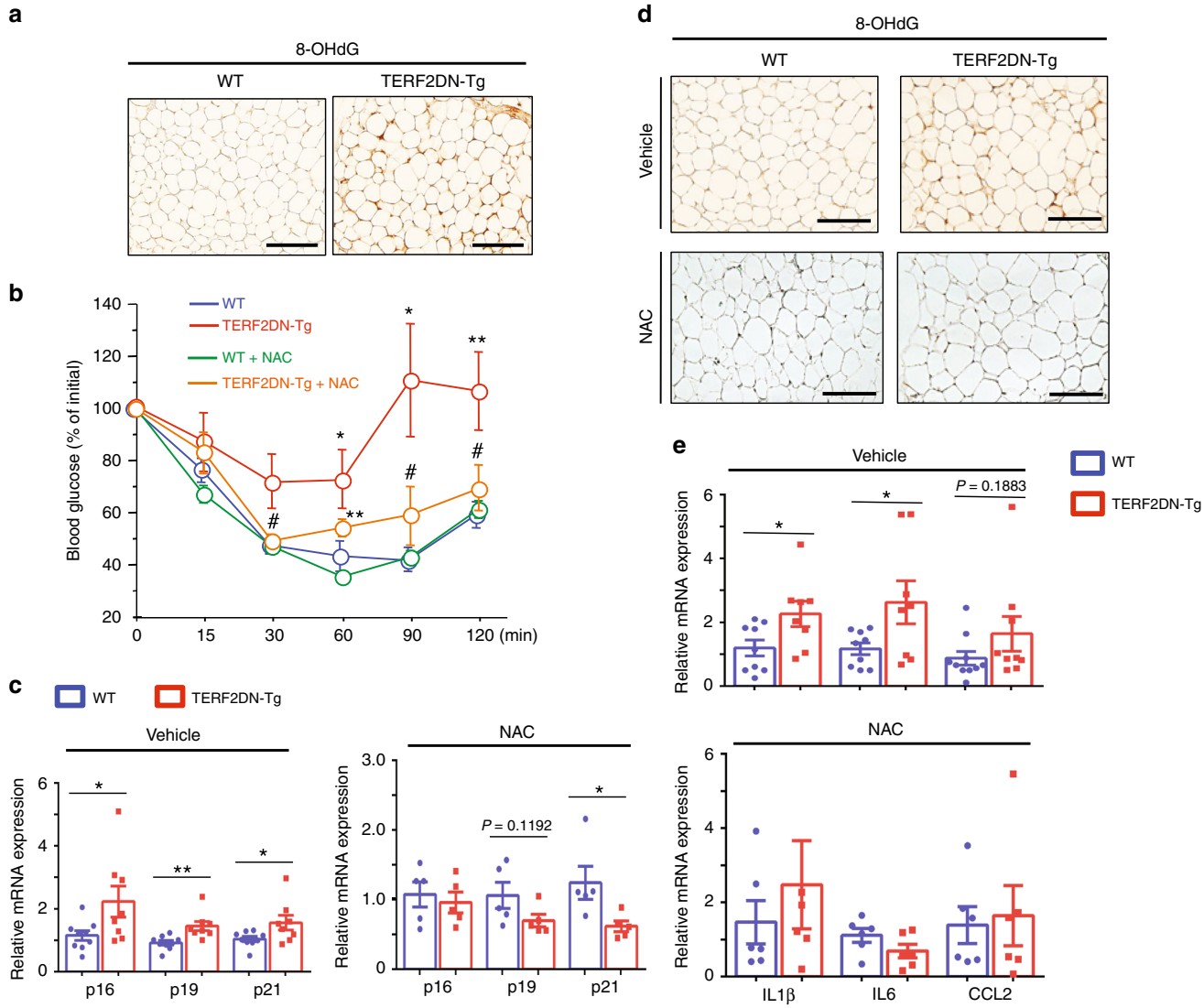

**Fig. 7 EC senescence impairs metabolic health by enhancing adipocyte oxidative stress. a** Immunostaining for the 8-OHdG in the WAT of WT or Tie2-TERF2DN-Tg mice at the age of 20 weeks old. **b** ITT in WT or Tie2-TERF2DN-Tg mice with or without NAC treatment for 10 weeks beginning at 10-week old ($n = 6$ biologically independent animals for WT; $n = 5$ biologically independent animals for Tg). **c** CDK inhibitor expression in the WAT of control (vehicle-treated, $n = 9$ biologically independent samples for WT; $n = 8$ biologically independent samples for Tg) or NAC-treated ($n = 5$ biologically independent samples each) WT or Tie2-TERF2DN-Tg mice. Non-repeated ANOVA with post hoc analysis of Fisher's PLSD was used for difference evaluation between the groups. **d** Immunostaining for the 8-OHdG in the WAT of WT or Tie2-TERF2DN-Tg mice treated with either vehicle or NAC. **e** Inflammatory gene expression in the WAT of control (vehicle-treated, $n = 9$ biologically independent samples for WT; $n = 8$ biologically independent samples for Tg) or NAC-treated ($n = 5$ biologically independent samples each) WT or Tie2-TERF2DN-Tg mice. A two-tailed Student's $t$ test was used for difference evaluation between the two groups (**c**, **e**). Data are presented as mean ± s.e. *$P < 0.05$, **$P < 0.01$, ***$P < 0.001$, ****$P < 0.0001$ and #not significant. Bars: 100 μm. Source data are provided as a Source Data file.

for age-related disease, and thus provides scientific evidence for the famous saying: "a man is as old as his arteries".

## Methods

**Cell culture**. HUVECs were regularly cultured in HuMedia-EG2 medium (Kurabo). Replicative senescent EC was prepared by culturing HUVEC for an extended period until passages 20–22. HUVECs in passages 4–5 were used as proliferating young control cells. Premature senescent EC was prepared by overexpressing DN form of TERF2 (TERF2DN)[25] in HUVECs at passages 4–5. The plasmid containing the TERF2-ΔB-ΔM (deletion mutant lacking the N-terminal basic domain and C-terminal Myb domain) was obtained from Addgene (plasmid #2431). For the transfection of TERF2DN, ECs were incubated with retrovirus carrying either DN TERF2 gene or GFP for 24 h, followed by incubation with fresh growth medium for 24–72 h before use for experiments. To prepare the conditioned medium (CM), ECs were given Dulbecco's modified Eagle medium (DMEM) supplemented with 5% FBS when ECs reached subconfluency. The conditioned medium was collected after 24 h of incubation. To prepare the control medium,

DMEM supplemented with 5% FBS was put in a culture dish without cells and incubated in the $CO_2$ incubator for 24 h in a parallel way.

3T3-L1 preadipocytes were obtained from Health Science Research Resources Bank (#JCRB9014). To prepare differentiated 3T3-L1 adipocytes, adipogenesis was induced as follows. Confluent 3T3-L1 preadipocytes were treated with 0.2 μM insulin, 0.25 μM dexamethasone, and 0.5 mM isobutylmethylxanthine at day 0 for 48 h, followed by the treatment with insulin (0.2 μM) for another 48 h. Afterward, cells were cultured in the DMEM supplemented with 10% FBS. For experiments, 3T3-L1 adipocytes at days 10–12 post differentiation were regularly used. For EC-SASP experiments, we regularly prepared a mixture of fresh growth medium and either of the control or CM derived from young or senescent EC at a 1:1 ratio, and cultured cells in this mixture for 4 days with changing the medium every other day. In some experiments, 3T3-L1 adipocytes were cultured in EC-CM containing 10 mM NAC (Sigma), 50 μM β-NMN (Oriental Bio), 10 μM NOX inhibitor III (Calbiochem), or 100 μM allopurinol (SelleckChem). For insulin signaling analysis, cells were incubated in serum-free medium for 3 h and then stimulated with 10 nM insulin, followed by protein extraction 20 min after insulin treatment.

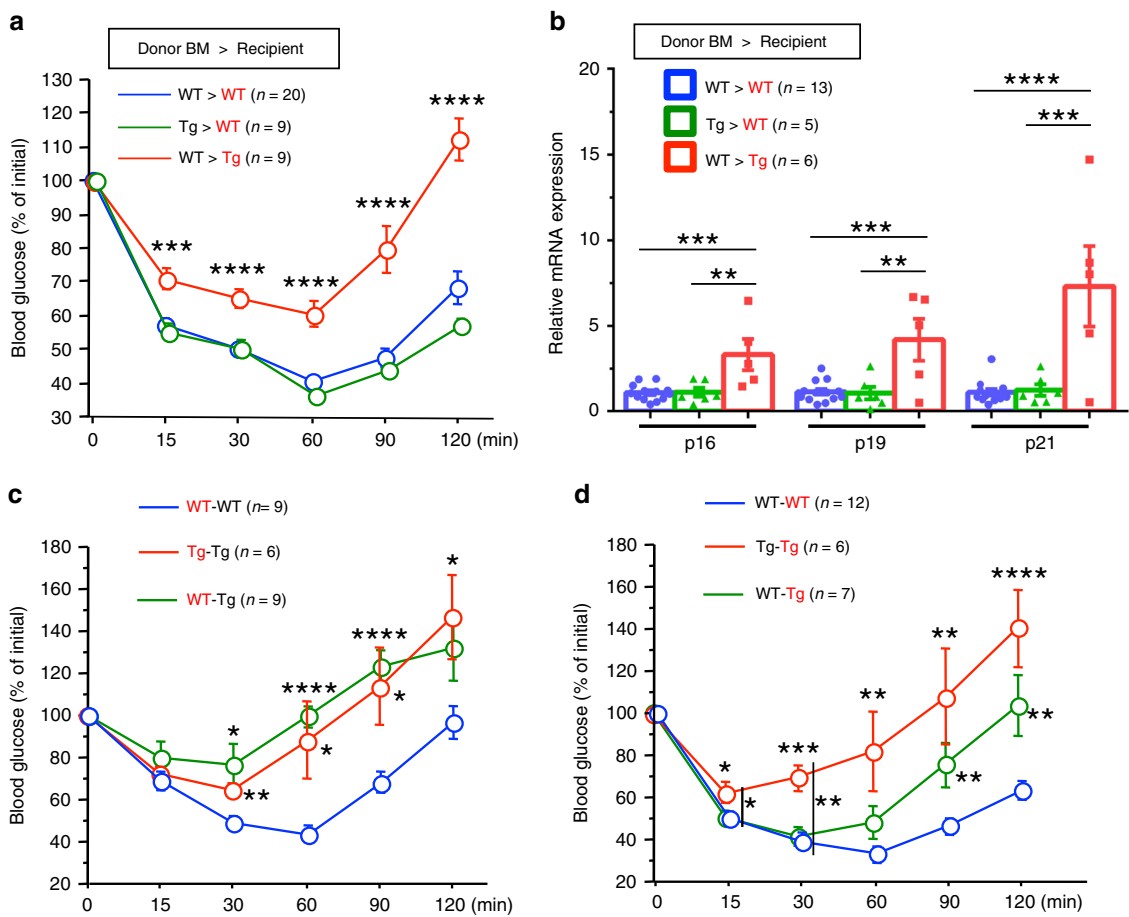

**Fig. 8 EC senescence impairs metabolic health through the SASP. a** ITT in recipient WT or Tie2-TERF2DN-Tg mice harboring BM of WT or Tg mice ($n =$ 20 biologically independent animals for WT>WT group; $n = 9$ biologically independent animals for Tg>WT group; $n = 9$ biologically independent animals for WT>Tg group). **b** CDK inhibitor expression in the WAT isolated from recipient WT or Tie2-TERF2DN-Tg mice harboring BM of WT or Tg mice ($n = 13$ biologically independent samples for WT>WT group; $n = 5$ biologically independent samples for Tg>WT group; $n = 6$ biologically independent samples for WT>Tg group). **c** ITT in the recipient WT or Tie2-TERF2DN-Tg mice whose circulation was shared with WT (recipient–donor; WT–WT, $n = 9$ biologically independent animals) or Tie2-TERF2DN-Tg mice ($n = 9$ biologically independent animals for WT-Tg group; $n = 6$ biologically independent animals for Tg–Tg group). **d** ITT in the donor WT or Tie2-TERF2DN-Tg mice whose circulation was shared with WT (WT–WT, $n = 12$ biologically independent animals) or Tie2-TERF2DN-Tg mice ($n = 7$ biologically independent animals for WT-Tg group; $n = 6$ biologically independent animals for Tg–Tg group). Non-repeated ANOVA with post hoc analysis of Fisher's PLSD was used for difference evaluation between the groups (**a–d**). Data are presented as mean ± s.e. *$P < 0.05$, **$P < 0.01$, ***$P < 0.001$, and ****$P < 0.0001$. Source data are provided as a Source Data file.

For the 3T3-L1 preadipocyte experiments, after the first day of plating, the growth medium was replaced with a mixture of fresh growth medium and either of the control or CM derived from young or senescent EC, and incubated for 4 days with changing the medium every other day. The C2C12 mouse skeletal myoblasts (ATCC #CRL-1772) were cultured in DMEM supplemented with 10% fetal bovine serum (FBS). Myogenesis was induced as follows. When C2C12 myoblasts reached 70–80% confluence, the growth medium was replaced with the differentiation medium (DMEM supplemented with 2% horse serum), and further incubated for 4 days. Fresh differentiation medium was given every other day. Afterward, differentiated C2C12 myotubes were cultured in a mixture of fresh growth medium, and either of the control or CM derived from young or senescent EC for 4 days with changing the medium every other day for EC-SASP experiments.

To activate inflammasomes in ECs, young and replicative senescent ECs were treated with urate crystal (MSU, 200 or 400 µg/mL) or adenosine triphosphate (ATP, 2.5 or 5 mM) overnight, and subsequently the culture medium was collected, followed by measurements of IL-1a by ELISA. In some experiments, young and replicative senescent ECs were treated with 500 ng/ml IL-1 receptor antagonist overnight. Also, in some experiments, premature senescent EC was transfected with 10 nM siRNA for human IL-1a by using lipofectamine RNAiMAX (Thermo).

**SA-β-Gal staining**. SA-β-Gal staining was performed as follows. Cells were fixed with 4% PFA, and then incubated with SA-β-Gal staining solution (40 mM sodium phosphate, pH 6.0 + 5 mM potassium ferrocyanide + 5 mM potassium ferricyanide + 150 mM NaCl + 2 mM MgCl₂ + 1 mg/ml X-Gal) at 37 °C. Cells were observed every 1 h during the first 6 h, and subsequently every 4–8 h. When the

cells are stained as blue–green visualized under an inverted bright-field microscope, the reaction was terminated by washing with pure water.

The WAT for SA-β-Gal staining was prepared by perfusion fixation with 4% PFA. The WAT was further fixed with 4% PFA for several days at 4 °C, and then washed with PBS. Fixed WAT was incubated in the SA-β-Gal staining solution at 37 °C until stained as blue–green appearance by observing the samples every 1 h.

**SPiDER-β-Gal staining**. Cells were fixed with 4% PFA, and then incubated with ~0.3 ng/µl SPiDER-β-Gal (DOJINDO) in McIlvaine buffer (pH 6.0) for 30 min at 37 °C. The nucleus was stained with Hoechst 33342. Following the wash with Hanks' Balanced Salt Solution, cells were observed under fluorescence microscopy.

**Animal study**. All experimental protocols were approved by the Ethics Review Committee for Animal Experimentation of Kobe Pharmaceutical University. All researchers have complied with all relevant ethical regulations for animal testing and research. Transgenic mice that overexpressed TERF2DN in EC (Tie2-TERF2DN-Tg) were generated (C57/BL6J background). The plasmid containing the TERF2-ΔB-ΔM was obtained from Addgene (plasmid #2431)[25]. The plasmid containing the Tie2 promoter and enhancer was a gift from Dr. Thomas N. Sato. The Tg mice were propagated as heterozygous Tg animals by breeding with WT C57/BL6J mice.

Mice were fed a normal chow diet (containing 23.1% protein and 5.1% fat) with ad libitum access to food and water.

For prevention experiments, 40 mM NAC in drinking water was given for 10 weeks beginning at the age of 10-week old. Subcutaneous and visceral

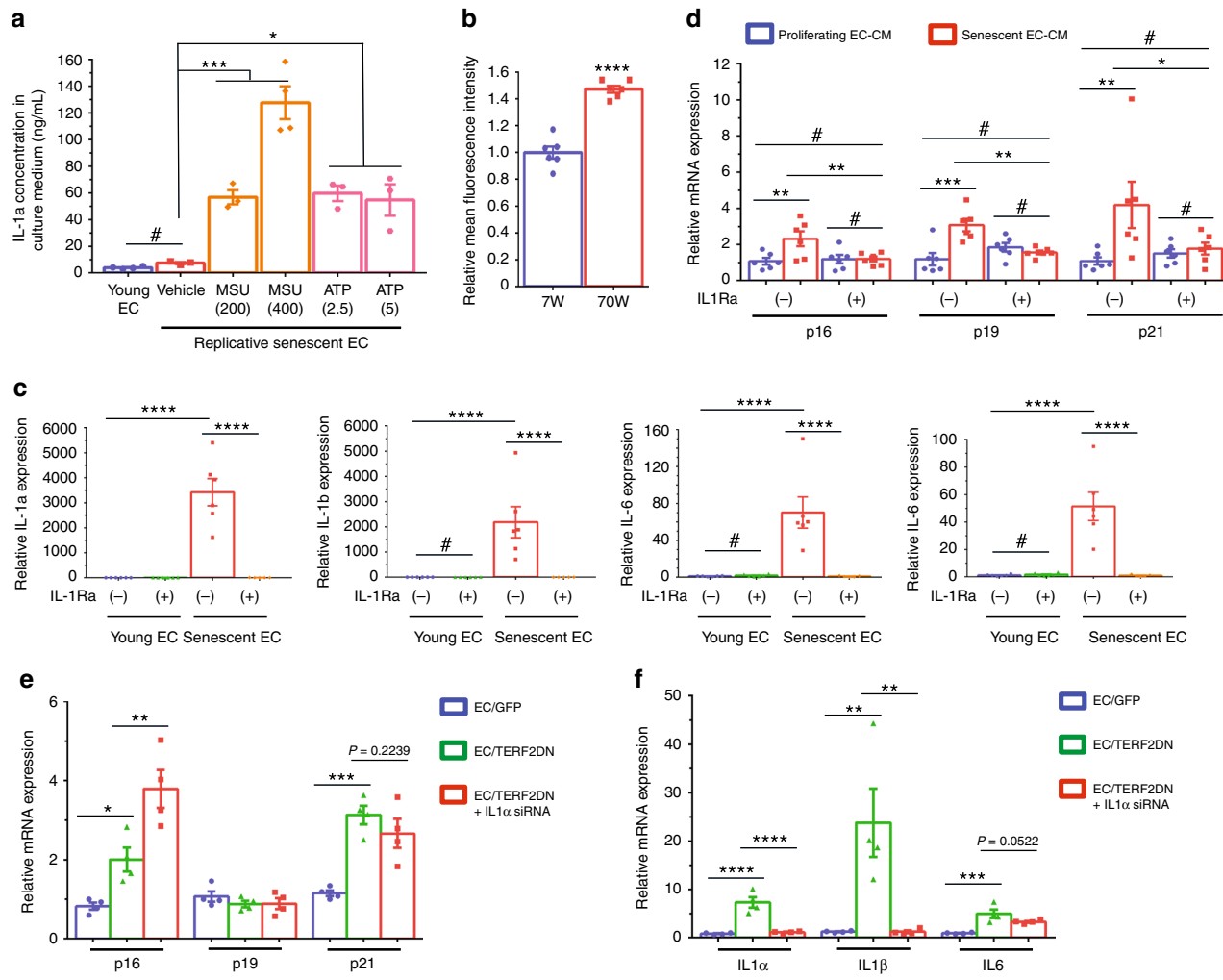

**Fig. 9 IL-1a orchestrates senescence-associated cytokine networks in EC. a** IL-1a concentration in culture medium derived from young or replicative senescent EC in the presence or absence of inflammasome activators, such as monosodium urate crystal (MSU, 200 or 400 μg/mL) and adenosine triphosphate (ATP, 2.5 or 5 mM) ($n = 4$ biologically independent samples for young; $n = 3$ biologically independent samples for senescent EC). **b** FACS analysis of cell-surface IL-1a in ECs isolated from the lung of young (7W) or aged (70W) mice ($n = 6$ biologically independent samples each). A two-tailed Student's $t$ test was used for difference evaluation between the two groups. **c** SASP factor expression in young or replicative senescent EC in the presence or absence of IL-1 receptor antagonist ($n = 6$ biologically independent samples each). **d** CDK inhibitor expression in 3T3-L1 adipocytes treated with the conditioned medium derived from young or replicative senescent EC with or without IL-1 receptor antagonist treatment ($n = 6$ biologically independent samples each). **e** CDK inhibitor expression in young control EC (GFP), premature senescent EC (TERF2DN), or premature senescent EC transfected with IL-1a siRNA ($n = 4$ biologically independent samples each). **f** SASP factor expression in young control EC (GFP), premature senescent EC (TERF2DN), or premature senescent EC transfected with IL-1a siRNA ($n = 4$ biologically independent samples each). Non-repeated ANOVA with post hoc analysis of Fisher's PLSD was used for difference evaluation between the groups (**a**, **c–f**). Data are presented as mean ± s.e. *$P < 0.05$, **$P < 0.01$, ***$P < 0.001$, ****$P < 0.0001$ and #not significant. Source data are provided as a Source Data file.

epididymal WAT were used for SA-β-Gal staining. Visceral epididymal WAT was used for RNA and protein extraction, and histological analysis. The interscapular BAT was used for RNA extraction and histological analysis. Liver, and soleus muscle were used for RNA and protein extraction. For RNA extraction, QIAzol lysis reagent and RNeasy Lipid Tissue Mini Kit (QIAGEN) were used for WAT and BAT; RNAiso Plus (Takara) and NucleoSpin RNA Clean-up Kit (MACHEREY-NAGEL) were used for liver and skeletal muscle. For protein extraction, tissues were homogenized in RIPA buffer containing protease and phosphatase inhibitors. For blood sampling, mice were fasted for 6 h and then blood was collected from the heart. After sampling, blood was incubated for 1 h at room temperature, and then centrifuged at $1500 \times g$ for 10 min, followed by serum collection. Serum insulin levels were measured by the ELISA.

**Parabiosis.** Shared circulation by parabiosis was performed by using the peritoneal method as follows. WT or Tie2-TERF2DN-Tg mice at 10-week old were used as recipient or donor mice. Mice were anesthetized by using inhalational isoflurane. The recipient and donor mice were then shaved on the left or right side, respectively. A U-shaped skin incision was created on the shaved side of one mouse, and

an inverse U-shaped skin incision was made on the shaved side of the other mouse. In both mice, the incision extended from the shoulder to the leg. These two mice were joined, side by side, by connecting skin in the elbow and knee joint area and some part of peritoneum and abdominal muscle by using nonabsorbable 4-0 sutures. Connected mice were kept on a heated pad (adjusted to 37 °C) until their complete recovery from anesthesia. The phenotypic analysis was performed after a 10-week period of shared circulation.

**Statistical analysis.** All data are presented as mean ± SEM. Differences between groups were analyzed by using two-tailed Student's $t$ test. Comparisons among more than three groups were assessed for significance by non-repeated ANOVA with post hoc analysis of Fisher's PLSD. $P < 0.05$ was considered statistically significant.

**Reporting summary.** Further information on research design is available in the Nature Research Reporting Summary linked to this article.

## Data availability

The authors declare that all data supporting the findings of this study are available within the paper and its Supplementary information files. The source data underlying Figs. 1b–f, 2c, 3c, 3e, 4a–d, 5b, 5c, 5g, 5h, 6b–d, 7c, 7e, 8b, and 9a–f are provided as the Source Data file. The mass spectrometry proteomics data have been deposited to the ProteomeXchange Consortium via the PRIDE[55] partner repository with the dataset identifier PXD016932 and 10.6019/PXD016932.

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

## Acknowledgements

This work was supported by JSPS KAKENHI Grant Numbers JP16K09524, JP19K08502, and Kobayashi International Scholarship Foundation. We thank Pranindya Rinastiti, Gusty Rizky Teguh Ryanto, Risa Ramadhiani, Andreas Haryono, and Adam Prabata for

their excellent technical assistance. We thank Dr. Eiji Hara for discussion and helpful suggestions. We thank Dr. Matthias Barton for critical reading of this paper.

## Author contributions

K.I. and N.E. conceived and designed research. A.J.B., D.B.N., K.I., D.A.W., R.U., S.H. and N.S. performed metabolic and molecular studies. K.I., S.M., K.-I.H. and N.E. wrote the paper.

## Competing interests

The authors declare no competing interests.
