## [Peer Review File · Nature Communications]

Reviewers' comments:

Reviewer #1 (Remarks to the Author):

Investigators present results of a comprehensive and thorough study of metabolic and functional aberrations of adipose tissue (AT) in a mouse model of premature endothelial cell senescence induced by telomeric dysfunction. Authors provide solid evidence to implicate SASP in impaired functions of AT. Several suggestions and comments are summarized below.

1. In general, it would have been so much more innovative to present results of proteomic/metabolomic screen of the conditioned media and identify, even tentatively, endothelial SASP factor(s) responsible for inducing senescence of AT.
2. The title refers to "systemic metabolic health" - this is an obscure term. In my view, authors describe the status of AT and its metabolic effects.
3. The authors interpret senescence of endothelial and AT in terms of a linear progression. This relationship may be more complicated. For instance, endothelial senescence leading to vascular rarefaction may per se induce oxidative stress, insulin resistance, and other abnormalities described in the manuscript, including the secondary AT senescence. Along these lines, it would be informative to trace the progression of endothelial and adipocyte abnormalities in time (rather than selecting a single time-point of 20 weeks). Also, does endothelial SASP have an autocrine effects?
4. It is somewhat surprising that other investigated tissues appeared to be "resistant" to endothelial SASP - could it be again the problem of timing?
5. Oxidative stress induced by SASP is studied somewhat superficially, although it is known that ROS can induce premature senescence even without attrition of telomeres.
6. Having determined glucose and insulin levels, it would be reasonable to calculate HOMA.

Michael S Goligorsky, MD, PhD

Reviewer #2 (Remarks to the Author):

This is an interesting manuscript to assess the effects of endothelial cell senescence on the function of adipocytes and adipose tissue. The authors base their experiments on the premise that adipose tissue vasculature plays a crucial role in adipose function, through control of oxygen and nutrient fluxes, as well as being the niche for adipose tissue progenitors. They perform in-vitro experiments to test whether senescent endothelial cells secrete factors that affect cultured adipocyte function and perform in-vivo experiments to determine whether endothelial cell senescence affects metabolic homeostasis through endocrine mechanisms. The data is presented clearly and is interesting, however the major conclusions are not fully supported because alternative explanations exist. Additional experiments would need to be done to strengthen the conclusion that the secretome of senescent endothelial cells directly causes metabolic dysfunction. Specific comments are:

1. The results indicate that senescent endothelial cells modify culture medium in a way that then negatively affects cultured adipocyte function. However, these experiments do not prove that the deleterious effects result from secretion of senescence-associated deleterious factors. It is also possible that senescent endothelial cells are consuming a component of the medium, e.g. a trace element, that is required for optimal 3T3-L1 adipocyte insulin responsiveness.
2. The data in 3T3-L1 adipocytes indicates a decreased level of IRS-1 in response to culture in media from senescent cells. What are the levels of IRS-1 in adipose tissue from the transgenic mice? Is this result replicated? If not, what is the relevance of the in-vitro 3T3-L1 data to the biology of the transgenic mice?
3. All the data indicating that the effects of antioxidants mitigating the effects of senescent medium lack appropriate controls for the actions of antioxidants alone.
4. The legend for Figure 6 is missing.
5. Insulin tolerance curves comparing wild-type and transgenic mice tend to diverge after more than 30 min; this pattern is more consistent with an increased counter-regulatory response than with true insulin resistance. High levels of glucocorticoids could account for this behavior and should be assessed.
6. The authors show that transgenic mice display higher basal insulin levels compared to wild-type. It has been shown that moderate hyperinsulinemia can induce insulin resistance (1, 2). Therefore, the conclusion that a senescent endothelial cells secretome is responsible for the observed phenotype is not warranted.

1. Kim B, McLean LL, Philip SS, and Feldman EL. Hyperinsulinemia induces insulin resistance in dorsal root ganglion neurons. *Endocrinology*. 2011;152(10):3638-47.

2. Liu HY, Cao SY, Hong T, Han J, Liu Z, and Cao W. Insulin is a stronger inducer of insulin resistance than hyperglycemia in mice with type 1 diabetes mellitus (T1DM). *J Biol Chem.* 2009;284(40):27090-100.

Reviewer #3 (Remarks to the Author):

NCOMMS-19-10705-T

" Endothelial cell-specific progeria impairs systemic metabolic health in mice through the senescence-associated secretory phenotype" by Barinda and co-workers.

This is a potentially interesting paper, illustrating the importance of endothelial cell (EC) senescence on aging-associated adipocyte dysfunction. Overall, data are for the most part solid. However, there are several limitations, which in my view preclude its publication in *Nature Communications*, at least in its present form. Most importantly, it remains unclear whether senescent EC provoke aging-associated adipocyte dysfunction through SASP. Moreover, it also remains unclear how the SASP integrated into the elevation of ROS levels in adipocytes. More work is therefore needed to make this paper suitable for publication in *Nature Communications*.

Specific concerns:

1. In Figs. 1 and 2, the authors showed that the treatment with conditioned medium of senescent EC provokes senescence-like phenotypes in adipocytes. However, it remains unclear whether or not this is mainly due to SASP factors secreted from senescent ECs. Since IL1-alpha is known to be an upstream regulator of SASP induction (Orjalo et al *PNAS* 2009) and the expression of IL1-alpha is significantly increased in senescent ECs (Extended Figs. 1f and 2c), the authors should examine whether the onset of senescence-like phenotypes is abolished if the experiments described in Figs. 1 and 2 were repeated using ECs lacking IL1-alpha.

2. Along similar lines, although the authors concluded that senescent ECs increased ROS levels and impairs insulin signalling in adipocytes through SASP (Fig. 3), it remains unclear whether this is

indeed through the SASP. Moreover, the underlying mechanisms also remain unclear. Therefore, I strongly feel that the authors should address these points experimentally.

3. As the authors mentioned in the discussion part, senescent cells reportedly secrete not only SASP factors but also various micro vesicles. It is therefore difficult to conclude that SASP factors in blood circulation impaired systemic metabolic health in recipient WT mice (Fig. 8). Thus, it would be more interesting if the authors could repeat this experiment using IL1-alpha knockout background mice.

Reviewer #4 (Remarks to the Author):

This manuscript describes the importance of the endothelial cell (EC) senescence in systemic metabolic disorder in vitro and in vivo. The authors created EC-specific progeroid mice overexpressing the dominant negative form of telomeric repeat-binding factor 2 under the control of the Tie2 promoter which induced premature senescence specifically in EC. EC-specific progeroid mice showed systemic metabolic disorder with white adipose tissue dysfunction even with normal diet.

The authors created an interesting mouse model that shows the vascular-specific senescence. This manuscript shows the potentially interesting phenomenon that EC-derived secreted protein could induce adipocyte senescence in vitro. However, this work lacks molecular mechanism to explain which soluble factor is important for induction of adipocyte senescence. Moreover, although the authors focus on the senescence of adipocytes and WAT and show the induction of CDK inhibitor expression in WAT in EC-specific progeroid mice, there are little data regarding the SASP factors expression in adipocytes and WAT which could be important for systemic metabolic disorder. This work lacks the molecular mechanism of how senescence of adipocytes provokes systemic metabolic disorder in vivo.

Specific concerns:

(1) In Figure 2c and 3c, the authors showed that the conditioned medium (CM) from TERF2DN-expressing EC induced the upregulation of CDKI and oxidative stress in adipocytes. However, these data lack which factor in EC-derived CM promoted cellular senescence. Therefore, the authors should at least show which protein were secreted in CM by a method such as protein array, and should suggest the key factors for the induction of adipocyte senescence.

(2) In Figure 5d and e, the authors showed that EC-specific progeroid mice underwent cellular senescence in WAT. In this experiment, the authors should show the weight of mice and the weight of WAT of each group to more clearly show the phenotype of mice.

(3) In Figure 7a, 8-OHdG was positively stained in the WAT of EC-specific progeroid mouse. The authors should explain why such excess oxidative stress was induced in WAT and why WAT was more prone to be highly sensitive to oxidative stress compared to other tissues.

(4) In Figure 7a, which cells were stained with 8-OHdG in WAT? In Figure 7a, stained areas seem to be crown-like structures in WAT of TERF2DK-Tg. Therefore, the authors should show whether or not macrophages were stained with 8-OHdG.

(5) In Extended data Fig 1c, the authors showed tri-methyl histone (H3K9me3) positivity in EC. However, H3K9me3 and SAHF are thought to be insufficient as senescence marker particularly in the stage of SASP (eg: Kosar M, et al. Cell Cycle. 2011;10:457-68). Recently, SASP was reported to be correlated with a decline of H3K9me2 (Takahashi et al. Mol Cell. 2012 ;45:123-31). Therefore, the authors should check whether H3K9me2 level was declined by Western blotting comparing with control samples instead of H3K9me3 positivity.

(6) In Figure 7d and e, the authors showed the cytokine expression in WAT after NAC treatment. The authors should compare the cytokine expression with NAC-untreated samples side by side.

(7) The authors judged the metabolic disorder by ITT measuring blood glucose, and this seems to be too focused. The authors should also show other tests such as the glucose tolerance test at the same time.

Responses to the Reviewers' comments:

Reviewer #1 (Remarks to the Author):

Investigators present results of a comprehensive and thorough study of metabolic and functional aberrations of adipose tissue (AT) in a mouse model of premature endothelial cell senescence induced by telomeric dysfunction. Authors provide solid evidence to implicate SASP in impaired functions of AT. Several suggestions and comments are summarized below.

Comment-1

In general, it would have been so much more innovative to present results of proteomic/metabolomic screen of the conditioned media and identify, even tentatively, endothelial SASP factor(s) responsible for inducing senescence of AT.

Response-1

Thank you for the comments. We sincerely agree on importance of the SASP factor(s) screening. To obtain some clues for identification of endothelial SASP factors responsible for inducing senescence of AT, we have performed the shotgun proteomics analysis using the conditioned media (CM) derived from young control and senescent endothelial cells. The results were shown in the Extended Data Fig. 12 and Fig. 13, and Extended Data Excel sheet in the revised manuscript. As shown in the Excel sheet, we could detect 1307 proteins in the CM. Among these proteins, 380 proteins were differentially expressed ($P < 0.1$) in the CM derived from senescent ECs relative to young control cells. We have shown the heat map for these 380 proteins after clustering in Extended Data Fig. 12. We then performed a pathway analysis for possible endothelial SASP factors that are increased in the CM from senescent EC, and identified several pathways that could be modified by the senescent EC secretomes (Extended Data Fig. 13). These include signaling pathway for IGF, FGF and PDGF, which have been involved in cellular senescence and oxidative stress. However, as we mentioned in the Discussion, significant number of proteins detected in the CM were not secreted but localized intracellularly. We presume that some of them might be actively secreted through vesicles, and some of them might be derived from dead cells. Furthermore, many cytokines including IL-1a, IL-1b, IL-6, IL-8, MCP-1, TNF-a, and TGF-b were not detected in the CM, probably because of technical limitations. With these limitations, it is difficult to determine SASP factor(s) responsible for inducing

senescence of AT at this point; however, we believe that these proteomics data give some clues to identify important SASP factors when combined with additional complementary data in the future.

Comment-2

The title refers to "systemic metabolic health" - this is an obscure term. In my view, authors describe the status of AT and its metabolic effects.

Response-2

Thank you for the comment. According to the comment, we have changed the title to "Endothelial cell-specific progeria induces senescence in adipose tissue and impairs systemic insulin sensitivity in mice through the senescence-associated secretory phenotype" in the revised manuscript.

Comment-3

The authors interpret senescence of endothelial and AT in terms of a linear progression. This relationship may be more complicated. For instance, endothelial senescence leading to vascular rarefaction may per se induce oxidative stress, insulin resistance, and other abnormalities described in the manuscript, including the secondary AT senescence. Along these lines, it would be informative to trace the progression of endothelial and adipocyte abnormalities in time (rather than selecting a single time-point of 20 weeks). Also, does endothelial SASP have an autocrine effects?

Response-3

Thank you for the comments. As shown in Fig. 6, neither AT senescence nor reduced systemic insulin sensitivity was detected, while endothelial senescence was readily detected in EC-specific progeroid mice at the age of 10-week old. Also, we did not detect the vascular rarefaction in AT of EC-specific progeroid mice even at the age of 20-week old (Extended Data Fig 8d and 8e). Therefore, we think that senescent ECs directly induce AT senescence in vivo as was suggested by the in vitro experiments.

According to the comments, we have newly analyzed possible autocrine effects of endothelial SASP in ECs. As shown in Extended Data Fig. 4, treatment with the CM derived from senescent EC did not induce senescence phenotype in ECs.

Therefore, it is unlikely that endothelial SASP have an autocrine effects with regard to cellular senescence induction.

Comment-4

It is somewhat surprising that other investigated tissues appeared to be "resistant" to endothelial SASP - could it be again the problem of timing?

Response-4

Thank you for the comments. To address the comments, we have analyzed senescence phenotypes in other tissues of EC-specific progeroid mice at the older age. We put the Figures for Reviewers at the end of this letter, so please see these Figures. As shown in the Fig. 3a for Reviewers, wild-type mice typically showed healthier appearance at the age of 60-week old when compared to EC-specific progeroid mice. Furthermore, age-related cardiomyocyte hypertrophy in the heart and glomerulosclerosis in the kidney were deteriorated in EC-specific progeroid mice in association with increased CDK inhibitors expression (Fig. 3b-g for Reviewers). These data strongly suggest that endothelial SASP accelerates the senescence-associated organ dysfunction in not only AT but also multiple tissues including the heart and kidney. Notably, these accelerated senescence phenotypes in the heart and kidney were not detected in EC-specific progeroid mice at the age of 20-week old (Fig. 1 and Fig. 2 for Reviewers). Therefore, it seems that the timing when the deleterious effects of EC senescence appear is different among tissues as the Reviewer suggested. Although these data are interesting and important, we think that it may not be appropriate to include these data in the current manuscript because we focus on a role of senescent EC in AT and metabolic functions in this manuscript. We would like to publish these data in the future manuscript if the Reviewers and the Editor agree with it.

Comment-5

Oxidative stress induced by SASP is studied somewhat superficially, although it is known that ROS can induce premature senescence even without attrition of telomeres.

Response-5

Thank you for the comment. We have performed additional experiments for the analysis of oxidative stress induced by SASP. Since imbalanced redox state causes oxidative stress, we first analyzed whether excess ROS generation contributes to the enhanced oxidative stress in adipocytes treated with senescent EC-CM. As shown in Extended Data Fig. 5a, pharmacological inhibition of xanthine oxidase or NADPH oxidase failed to cancel the adipocyte senescence induced by endothelial SASP. In contrast, we have detected a significant reduction of superoxide dismutase expression in adipocytes treated with senescent EC-CM (Extended Data Fig. 5b). These data suggest that impaired elimination, rather than increased generation, of ROS might play an important role in the enhanced oxidative stress induced by endothelial SASP in adipocytes.

Comment-6

Having determined glucose and insulin levels, it would be reasonable to calculate HOMA.

Response-6

Thank you for the comment. We agree with the comment; however, the blood glucose and serum insulin levels were analyzed at a different timing. Moreover, blood glucose levels were analyzed without fasting, while the serum insulin levels were analyzed in mice after fasting. Therefore, we could not use these data to calculate HOMA.

Michael S Goligorsky, MD, PhD

Reviewer #2 (Remarks to the Author):

This is an interesting manuscript to assess the effects of endothelial cell senescence on the function of adipocytes and adipose tissue. The authors base their experiments on the premise that adipose tissue vasculature plays a crucial role in adipose function, through control of oxygen and nutrient fluxes, as well as being the niche for adipose tissue progenitors. They perform in-vitro experiments to test whether senescent endothelial cells secrete factors that affect cultured adipocyte function and perform in-vivo experiments to determine whether endothelial cell senescence affects metabolic homeostasis through endocrine mechanisms. The data is presented clearly and is interesting, however the major conclusions are not fully supported because alternative explanations exist. Additional experiments would need to be done to strengthen the conclusion that the secretome of senescent endothelial cells directly causes metabolic dysfunction. Specific comments are:

Comment-1

The results indicate that senescent endothelial cells modify culture medium in a way that then negatively affects cultured adipocyte function. However, these experiments do not prove that the deleterious effects result from secretion of senescence-associated deleterious factors. It is also possible that senescent endothelial cells are consuming a component of the medium, e.g. a trace element, that is required for optimal 3T3-L1 adipocyte insulin responsiveness.

Response-1

Thank you for the comments. We agree that there is a possibility that enhanced consumption of elements by senescent EC may cause the dysfunction of adipocytes treated with conditioned media derived from senescent EC. However, we think it is unlikely because of several reasons; 1. When treat cells with conditioned media, we always prepared the mixture of fresh growth medium and conditioned media at 1:1 ratio, and cultured cells in the freshly prepared mixture medium. We described this point clearly in the Method of the revised manuscript. Therefore, none of elements were depleted in the culture media although some elements could decrease down to half; 2. We have performed new experiments to assess a role of IL-1 α , which have been reported to orchestrate the SASP in fibroblast, in senescence-associated endothelial secretomes. As shown in new Fig. 9, membrane-bound IL-1 α plays a critical role in endothelial SASP as well. The important finding was that conditioned media derived from senescent ECs treated with IL-1 receptor antagonist failed to induce adipocyte senescence in association with the reduction in SASP factors expression (Fig. 9d). These data further support the critical role of SASP in adipocyte dysfunction due to EC senescence. However, we have described a possibility that enhanced consumption of trace elements by

senescent EC could also be involved in the adipocyte dysfunction due to EC senescence in the Discussion of the revised manuscript.

Comment-2

The data in 3T3-L1 adipocytes indicates a decreased level of IRS-1 in response to culture in media from senescent cells. What are the levels of IRS-1 in adipose tissue from the transgenic mice? Is this result replicated? If not, what is the relevance of the in-vitro 3T3-L1 data to the biology of the transgenic mice?

Response-2

Thank you for the comments. We have newly analyzed the levels of IRS-1 in adipose tissue from WT and the transgenic mice. As shown in new Fig. 5g, IRS-1 levels were significantly reduced in the transgenic mice comparing to those in WT mice in consistent with the in-vitro 3T3-L1 data. These data further support our hypothesis that reduction of IRS-1 is causally involved in adipocyte dysfunction mediated by EC senescence.

Comment-3

All the data indicating that the effects of antioxidants mitigating the effects of senescent medium lack appropriate controls for the actions of antioxidants alone.

Response-3

The graph in Fig. 3c showed CDKI expressions in ECs relative to control ECs transfected with GFP and treated with vehicle. When focused on control ECs transfected with GFP, the data indicate that treatment with neither bNMN nor NAC affected the CDKI expression in EC. Therefore, antioxidants alone seem not to have significant actions on the CDKI expression in young control ECs.

Comment-4

The legend for Figure 6 is missing.

Response-4

We appreciate the comment. We put the legend for Fig. 6 in the revised manuscript.

Comment-5

Insulin tolerance curves comparing wild-type and transgenic mice tend to diverge after more than 30 min; this pattern is more consistent with an increased

counter-regulatory response than with true insulin resistance. High levels of glucocorticoids could account for this behavior and should be assessed.

Response-5

Thank you for the comment. According to the comment, we newly measured the serum corticosterone levels in WT and transgenic mice. As shown in Extended Data Fig. 7c, there was no difference in serum corticosterone levels between WT and transgenic mice at the age of 20-week old.

Comment-6

The authors show that transgenic mice display higher basal insulin levels compared to wild-type. It has been shown that moderate hyperinsulinemia can induce insulin resistance (1, 2). Therefore, the conclusion that a senescent endothelial cells secretome is responsible for the observed phenotype is not warranted.

1. Kim B, McLean LL, Philip SS, and Feldman EL. Hyperinsulinemia induces insulin resistance in dorsal root ganglion neurons. *Endocrinology*. 2011;152(10):3638-47.
2. Liu HY, Cao SY, Hong T, Han J, Liu Z, and Cao W. Insulin is a stronger inducer of insulin resistance than hyperglycemia in mice with type 1 diabetes mellitus (T1DM). *J Biol Chem*. 2009;284(40):27090-100.

Response-6

Thank you for the comments. We agree that hyperinsulinemia itself deteriorates insulin resistance in the advanced phase of metabolic disorders. However, insulin resistance should be in advance to the increase in insulin levels, and insulin resistance is the cause but not the result of hyperinsulinemia at the early phase of metabolic disorders. Insulin levels are tightly regulated, and insulin levels do not change independently of blood glucose levels unless pancreatic beta cells produce insulin disorderly due to insulinoma. Furthermore, insulin levels used in the referenced papers were relatively high (2-200 nM corresponds to 11.62-1162 ng/mL) compared to serum insulin levels in our mice (~0.225 ng/mL in WT and ~0.275 ng/mL in Tg mice). Therefore, we think that higher basal insulin levels in transgenic mice are the results of insulin resistance that was mediated by EC senescence, and presume that this modestly increased serum insulin levels may affect systemic insulin sensitivity minimally.

Reviewer #3 (Remarks to the Author):

NCOMMS-19-10705-T

" Endothelial cell-specific progeria impairs systemic metabolic health in mice through the senescence-associated secretory phenotype" by Barinda and co-workers.

This is a potentially interesting paper, illustrating the importance of endothelial cell (EC) senescence on aging-associated adipocyte dysfunction. Overall, data are for the most part solid. However, there are several limitations, which in my view preclude its publication in Nature Communications, at least in its present form. Most importantly, it remains unclear whether senescent EC provoke aging-associated adipocyte dysfunction through SASP. Moreover, it also remains unclear how the SASP integrated into the elevation of ROS levels in adipocytes. More work is therefore needed to make this paper suitable for publication in Nature Communications.

Specific concerns:

Comment-1

In Figs. 1 and 2, the authors showed that the treatment with conditioned medium of senescent EC provokes senescence-like phenotypes in adipocytes. However, it remains unclear whether or not this

is mainly due to SASP factors secreted from senescent ECs. Since IL1-alpha is known to be an upstream regulator of SASP induction (Orjalo et al PNAS 2009) and the expression of IL1-alpha is significantly increased in senescent ECs (Extended Figs. 1f and 2c), the authors should examine whether the onset of senesce-like phenotypes is abolished if the experiments described in Figs.1 and 2 were repeated using ECs lacking IL1-alpha.

Response-1

Thank you for the comments. According to the comments, we have newly performed experiments to analyze a role of IL-1a in endothelial SASP. We first measured the IL-1a levels secreted into conditioned media; however IL-1a was hardly detectable in the culture medium of replicative senescent EC despite the high IL-1a mRNA expression levels (new Fig. 9a). We then treated senescent EC with inflammasome activators, and found that IL-1a was abundantly secreted into the culture medium of replicative senescent ECs with these stimuli (new Fig. 9a). These data suggest that most of IL-1a expressed in senescent EC is membrane-anchored. We also found that ECs isolated from aged mice expressed higher levels of membrane-bound IL-1a compared to ECs from young mice as assessed by FACS analysis (new Fig. 9b). Because cell surface-bound IL-1a has been reported to

regulate senescence-associated cytokine networks as the Reviewer suggested, we explored the role of membrane-bound IL-1a in the endothelial SASP. Treatment with IL-1 receptor antagonist abolished the increase of multiple cytokines in senescent EC compared with those in young EC (new Fig. 9c). Of note, conditioned media derived from senescent EC that was treated with IL-1 receptor antagonist failed to induce senescence-like state in 3T3-L1 adipocytes (new Fig. 9d). Since IL-1 receptor antagonist inhibits both IL-1a and IL-1b, we further analyze an important role of IL-1a using IL-1a-specific siRNA. As shown in Fig. 9e and 9f, knockdown of IL-1a using siRNA abolished the increase of multiple cytokines in senescent EC without affecting the CDKI expression levels. These data revealed an important role of IL-1a in orchestrating the senescence-associated cytokine networks in ECs, and collectively suggest a critical role of endothelial SASP in adipocyte dysfunction due to EC senescence.

Comment-2

Along similar lines, although the authors concluded that senescent ECs increased ROS levels and impairs insulin signalling in adipocytes through SASP (Fig. 3), it remains unclear whether this is indeed through the SASP. Moreover, the underlying mechanisms also remain unclear. Therefore, I strongly feel that the authors should address these points experimentally.

Response-2

Thank you for the comments. As mentioned in the Response-1, treatment with IL-1 receptor antagonist abolished the increase of multiple cytokines in senescent EC, and notably abrogated the senescence induction in adipocytes. These data strongly suggest that senescent EC induced adipocyte dysfunctions through the SASP.

Also, we performed additional experiments for the analysis of oxidative stress induced by endothelial SASP. Since imbalanced redox state causes oxidative stress, we first analyzed whether excess ROS generation contributes to the enhanced oxidative stress in adipocytes treated with senescent EC-CM. As shown in Extended Data Fig. 5a, pharmacological inhibition of xanthine oxidase or NADPH oxidase failed to cancel the adipocyte senescence induced by endothelial SASP. In contrast, we have detected a significant reduction of superoxide dismutase expression in adipocytes treated with senescent EC-CM (Extended Data Fig. 5b). These data suggest that impaired elimination, rather than increased generation, of ROS might play an important role in the enhanced oxidative stress induced by endothelial SASP in adipocytes.

Comment-3

As the authors mentioned in the discussion part, senescent cells reportedly secrete not only SASP factors but also various micro vesicles. It is therefore difficult to conclude that SASP factors in blood circulation impaired systemic metabolic health in recipient WT mice (Fig. 8). Thus, it would be more interesting if the authors could repeat this experiment using IL1-alpha knockout background mice.

Response-3

Thank you for the comments. In a broad sense, extracellular vesicles are also included in the SASP (Nat Commun 8:15728, 2017; Int J Mol Sci, 17(9), 2016; Mol Aspects Med 60, 2018; Cell Rep 27, 2019). Therefore, the parabiosis experiments strongly suggest that EC senescence impaired systemic metabolic health through the SASP, although causative factors in blood circulation remain to be identified.

Furthermore, it remains unclear whether these causative factors are canonical SASP factors including cytokines or non-canonical various micro vesicles, as the Reviewer suggested. Thanks to the

Reviewer's comments, we could demonstrate a crucial role of IL-1a in the endothelial SASP-mediated adipocyte dysfunction in vitro in the revised manuscript as mentioned above. Therefore, we sincerely agree that experiments using IL-1a knockout background mice will provide invaluable information to elucidate a role of canonical SASP factors in vivo. We described this important issue in the Discussion of the revised manuscript, and we would like to perform these experiments in the future to further characterize the endothelial SASP.

Reviewer #4 (Remarks to the Author):

This manuscript describes the importance of the endothelial cell (EC) senescence in systemic metabolic disorder in vitro and in vivo. The authors created EC-specific progeroid mice overexpressing the dominant negative form of telomeric repeat-binding factor 2 under the control of the Tie2 promoter which induced premature senescence specifically in EC. EC-specific progeroid mice showed systemic metabolic disorder with white adipose tissue dysfunction even with normal diet.

The authors created an interesting mouse model that shows the vascular-specific senescence. This manuscript shows the potentially interesting phenomenon that EC-derived secreted protein could induce adipocyte senescence in vitro. However, this work lacks molecular mechanism to explain which soluble factor is important for induction of adipocyte senescence. Moreover, although the authors focus on the senescence of adipocytes and WAT and show the induction of CDK inhibitor expression in WAT in EC-specific progeroid mice, there are little data regarding the SASP factors expression in adipocytes and WAT which could be important for systemic metabolic disorder. This work lacks the molecular mechanism of how senescence of adipocytes provokes systemic metabolic disorder in vivo.

Specific concerns:

Comment-1

In Figure 2c and 3c, the authors showed that the conditioned medium (CM) from TERF2DN-expressing EC induced the upregulation of CDKI and oxidative stress in adipocytes. However, these data lack which factor in EC-derived CM promoted cellular senescence. Therefore, the authors should at least show which protein were secreted in CM by a method such as protein array, and should suggest the key factors for the induction of adipocyte senescence.

Response-1

Thank you for the comments. We sincerely agree on the importance of the SASP factor(s) screening. To obtain some clues for identification of endothelial SASP factors responsible for inducing adipocyte senescence, we have performed the shotgun proteomics analysis using the conditioned media (CM) derived from young control and senescent endothelial cells. The results were shown in the Extended Data Fig. 12 and Fig. 13, and Extended Data Excel sheet in the revised manuscript. As shown in the Excel sheet, we could detect 1307 proteins in the CM. Among these proteins, 380 proteins were differentially expressed ($P < 0.1$) in the CM derived from senescent ECs relative to young control cells. We have shown the heat map for these 380 proteins after clustering in Extended Data Fig. 12. We then

performed a pathway analysis for possible endothelial SASP factors that are increased in the CM from senescent EC, and identified several pathways that could be modified by the senescent EC secretomes (Extended Data Fig. 13). These include signaling pathways for IGF, FGF and PDGF, which have been involved in cellular senescence and oxidative stress. However, as we mentioned in the Discussion, a significant number of proteins detected in the CM were not secreted but localized intracellularly. We presume that some of them might be actively secreted through vesicles, and some of them might be derived from dead cells. Furthermore, many cytokines including IL-1a, IL-1b, IL-6, IL-8, MCP-1, TNF-a, and TGF-b were not detected in the CM, probably because of technical limitations. With these limitations, it is difficult to determine SASP factor(s) responsible for inducing adipocyte senescence at this point; however, we believe that these proteomics data give some clues to identify important SASP factors when combined with additional complementary data in the future.

Comment-2

In Figure 5d and e, the authors showed that EC-specific progeroid mice underwent cellular senescence in WAT. In this experiment, the authors should show the weight of mice and the weight of WAT of each group to more clearly show the phenotype of mice.

Response-2

Thank you for the comments. According to the comments, we have measured the weight of mice, and analyzed the body fat ratio of these mice. As shown in the new Fig. 5b and 5c, body weight and

body fat ratio in EC-specific progeroid mice showed tendency toward increase as compared to those in WT mice, although the differences did not reach statistical significance.

Comment-3

In Figure 7a, 8-OHdG was positively stained in the WAT of EC-specific progeroid mouse. The authors should explain why such excess oxidative stress was induced in WAT and why WAT was more prone to be highly sensitive to oxidative stress compared to other tissues.

Response-3

Thank you for the comment. In the revised manuscript, we have performed additional experiments for the analysis of oxidative stress induced by SASP *in vitro*. Since imbalanced redox state causes oxidative stress, we first analyzed whether excess ROS generation contributes to the enhanced oxidative stress in adipocytes treated with senescent EC-CM. As shown in Extended Data Fig. 5a,

pharmacological inhibition of xanthine oxidase or NADPH oxidase failed to cancel the adipocyte senescence induced by endothelial SASP. In contrast, we have detected a significant reduction of superoxide dismutase expression in adipocytes treated with senescent EC-CM (Extended Data Fig. 5b). These data suggest that impaired elimination, rather than increased generation, of ROS might play an important role in the enhanced oxidative stress induced by endothelial SASP in adipocytes. However, further analysis including the regulation of antioxidative genes through endothelial SASP is required to elucidate the detailed mechanism by which excess oxidative stress is induced in WAT of EC-specific progeroid mice.

As the Reviewer suggested, our *in vitro* and *in vivo* data suggested that adipocytes are prone to be highly sensitive to endothelial SASP. It remains unclear whether adipocytes are more sensitive to oxidative stress or oxidative stress is enhanced more severely in adipocytes than in other types of cells by EC senescence.

As mentioned in the Response-4 for Reviewer #1, we have newly analyzed senescence phenotypes in other tissues of EC-specific progeroid mice at the older age. We have put the Figures for Reviewers at the end of these responses. Please see these Figures. As shown in the Fig. 3a for Reviewers, wild-type mice typically showed healthier appearance at the age of 60-week old when compared to

EC-progeroid mice. Furthermore, age-related cardioomyocyte hypertrophy in the heart and glomerulosclerosis in the kidney were deteriorated in EC-progeroid mice in association with increased CDK inhibitor expressions (Fig. 3b-g for Reviewers).

These data strongly suggest that endothelial SASP affect the

senescence-associated organ dysfunction in not only WAT but also multiple tissues including the heart and kidney. Notably, these accelerated senescence phenotype in the heart and kidney was not detected in EC-specific progeroid mice at the age of 20-week old (Fig. 1 and Fig. 2 for Reviewers). Therefore, it is still suggested that WAT is more sensitive to EC senescence than other tissues. At this point, we do not have sufficient data to explain why WAT is more prone to be highly sensitive to EC senescence, and further investigation is needed to elucidate this important issue as we discussed in the revised manuscript.

Comment-4

In Figure 7a, which cells were stained with 8-OHdG in WAT? In Figure 7a, stained areas seem to be crown-like structures in WAT of TERF2DK-Tg. Therefore, the authors should show whether or not macrophages were stained with 8-OHdG.

Response-4

Thank you for the comment. Because these mice were fed with normal chow, these mice were lean but not obese although body weight and body fat ratio showed tendency toward increase in EC-specific progeroid mice comparing to those in WT

mice. Therefore, crown-like structures were hardly detectable in the WAT of these mice, and the number of macrophages in the WAT was comparable between WT and EC-specific progeroid mice (Fig. 4a and 4b for Reviewers supplemented with this letter). When performed double staining for macrophage marker and 8-OHdG, 8-OHdG staining was mostly detected in adipocytes of EC-specific progeroid mice, but some of macrophages were also positive for 8-OHdG even in WT mice (Fig. 4c for Reviewers). There is a report describing that macrophages accumulated in atherosclerotic plaque were positive for 8-OHdG (Circ Res 114(3), 2014), but we do not know if macrophages in the WAT suffer from oxidative DNA damage even in lean mice. It appeared that the number of 8-OHdG-positive macrophages was similar between WT and EC-specific progeroid mice, and 8-OHdG-positive signals were mostly derived from adipocytes. Although it is interesting that some of macrophages in

WAT are positive for 8-OHdG, because these issues are out of focus of this manuscript, we did not include these data in the revised manuscript.

Comment-5

In Extended data Fig 1c, the authors showed tri-methyl histone (H3K9me3) positivity in EC. However, H3K9me3 and SAHF are thought to be insufficient as senescence marker particularly in the stage of SASP (eg: Kosar M, et al. Cell Cycle. 2011;10:457-68). Recently, SASP was reported to be correlated with a decline of H3K9me2 (Takahashi et al. Mol Cell. 2012 ;45:123-31). Therefore, the authors should check whether H3K9me2 level was declined by Western blotting comparing with control samples instead of H3K9me3 positivity.

Response-5

Thank you for the comment. According to the comment, we have checked the H3K9me2 level by Western blotting. As shown in Extended Data Fig. 1d, H3K9me2 protein levels were declined in senescent ECs as compared to those in young control cells.

Comment-6

In Figure 7d and e, the authors showed the cytokine expression in WAT after NAC treatment. The authors should compare the cytokine expression with NAC-untreated samples side by side.

Response-6

Thank you for the comment. According to the comment, we put the data of NAC-untreated samples side by side as shown in new Fig. 7d and 7e.

Comment-7

The authors judged the metabolic disorder by ITT measuring blood glucose, and this seems to be too focused. The authors should also show other tests such as the glucose tolerance test at the same time.

Response-7

Thank you for the comments. As shown in Extended Data Fig. 6a, glucose tolerance test showed minimal changes in EC-specific progeroid mice despite the significant difference in insulin sensitivity. This is probably because higher insulin secretion from pancreas could usually compensate the glucose intolerance in mild insulin resistance.

Regarding the molecular mechanism of how senescence of adipocytes provokes systemic metabolic disorder (which was pointed out in the general comment of this Reviewer), we presume that impaired insulin signaling in adipocytes plays a primary role in the systemic metabolic disorders observed in EC-specific progeroid mice. We have revealed that insulin signaling in adipocytes was impaired by senescent ECs in association with adipocyte senescence and IRS-1 reduction both in vitro and in vivo. Many previous papers reported a critical role of WAT insulin signaling in the regulation of systemic metabolic homeostasis (Nat Rev Mol Cell Biol 9(5):367-377, 2008; J Clin Invest 106(4):473-481, 2000). Furthermore, it has been reported that enhancing insulin signaling in adipocytes is sufficient to improve systemic metabolic homeostasis, while impaired insulin signaling in adipocytes causes systemic metabolic disorders (Nat Commun 6:7906, 2015; Proc Natl Acad Sci U S A. 115(7):1529-1534, 2018). Therefore, adipocyte senescence provokes systemic metabolic disorders in vivo at least partially through the impaired insulin signaling in WAT. We described these issues in the Discussion of the revised manuscript.

[redacted]

REVIEWERS' COMMENTS:

Reviewer #1 (Remarks to the Author):

I'm satisfied with extensive revisions presented in the revised manuscript.

Reviewer #2 (Remarks to the Author):

The authors have addressed all concerns raised in the initial review.

Reviewer #3 (Remarks to the Author):

NCOMMS-19-10705A

" Endothelial cell-specific progeria induces senescence in adipose tissue and impairs systemic insulin sensitivity in mice through the senescence-associated secretory phenotype " by Barinda and co-workers.

I have read revised manuscript and responses to the referee's comments, and found that all points raised by me were adequately addressed. Therefore, I think the manuscript is now appropriate for publication in Nature Communications.

Reviewer #4 (Remarks to the Author):

I have read the revised manuscript and the responses to the referees' comments, and found that almost all the points raised by referees were adequately addressed, except from IL-1 α deficient mouse experiment. However, importance of IL-1 α has been clearly shown in the experiments using cultured cells. Therefore, the additional data in the revised manuscript efficiently strengthen the significance and the mechanism of adipocyte dysfunction by vascular senescence-associated factors mediating inter-tissue crosstalk.

Responses to the Reviewers' comments

Reviewer #1 (Remarks to the Author):

I'm satisfied with extensive revisions presented in the revised manuscript.

Reviewer #2 (Remarks to the Author):

The authors have addressed all concerns raised in the initial review.

Reviewer #3 (Remarks to the Author):

NCOMMS-19-10705A

" Endothelial cell-specific progeria induces senescence in adipose tissue and impairs systemic insulin sensitivity in mice through the senescence-associated secretory phenotype " by Barinda and co-workers.

I have read revised manuscript and responses to the referee's comments, and found that all points raised by me were adequately addressed. Therefore, I think the manuscript is now appropriate for publication in Nature Communications.

Reviewer #4 (Remarks to the Author):

I have read the revised manuscript and the responses to the referees' comments, and found that almost all the points raised by referees were adequately addressed, except from IL-1 α deficient mouse experiment. However, importance of IL-1 α has been clearly shown in the experiments using cultured cells. Therefore, the additional data in the revised manuscript efficiently strengthen the significance and the mechanism of adipocyte dysfunction by vascular senescence-associated factors mediating inter-tissue crosstalk.

Response: We sincerely appreciate the Reviewers' positive evaluation on our revised manuscript.